# A polychromatic 'greenbeard' locus determines patterns of cooperation in a social amoeba

Nicole Gruenheit[1], Katie Parkinson[1], Balint Stewart[1], Jennifer A. Howie[1], Jason B. Wolf[2]
& Christopher R.L. Thompson[1]

Cheaters disrupt cooperation by reaping the benefits without paying their fair share of associated costs. Cheater impact can be diminished if cooperators display a tag ('greenbeard') and recognise and preferentially direct cooperation towards other tag carriers. Despite its popular appeal, the feasibility of such greenbeards has been questioned because the complex patterns of partner-specific cooperative behaviours seen in nature require greenbeards to come in different colours. Here we show that a locus ('Tgr') of a social amoeba represents a polychromatic greenbeard. Patterns of natural Tgr locus sequence polymorphisms predict partner-specific patterns of cooperation by underlying variation in partner-specific protein–protein binding strength and recognition specificity. Finally, Tgr locus polymorphisms increase fitness because they help avoid potential costs of cooperating with incompatible partners. These results suggest that a polychromatic greenbeard can provide a key mechanism for the evolutionary maintenance of cooperation.

[1] Faculty of Biology, Medicine and Health, Department of Developmental Biology and Medicine, The University of Manchester, Michael Smith Building, Oxford Road, Manchester M13 9PT, UK. [2] Milner Centre for Evolution and Department of Biology and Biochemistry, University of Bath, Claverton Down, Bath BA2 7AY, UK. Correspondence and requests for materials should be addressed to J.B.W. (email: jason@evolutionarygenetics.org) or to C.R.L.T. (email: christopher.thompson@manchester.ac.uk).

The evolutionary maintenance of cooperation is a conundrum because cooperative systems are under constant threat from selfish individuals that reap the fitness benefits of cooperative investments by others without paying their fair share of the associated costs. However, cooperation can be stabilized against such exploitation by mechanisms that allow cooperators to direct behaviour towards cooperative partners and avoid cheaters. For example, when relatives cooperate, their shared ancestry can create such an association and cooperation can be stabilized by kin selection[1–4]. However, while kinship facilitates cooperation by establishing a genetic association between interactants on average, signals that allow individuals to directly assess the genotype of potential partners at a gene (or tightly linked gene locus) that governs cooperation can provide more accurate information, and hence lead to more stable cooperation[1]. Hamilton[1] postulated a scenario in which a gene or locus ('supergene') has three properties. First, the locus results in the display of some phenotypic marker. Second, the locus determines the perception of the marker[1,5,6]. Third, individuals use the perception of the marker to modulate their social response (engagement in cooperation) towards other carriers of the locus (including acting selfishly towards non carriers). Dawkins[6] illustrated this concept using an abstract scenario in which behavioural variation is discrete (there are altruists and non-altruists) and governed by a gene that produces the recognizable characteristic of a green beard, that individuals with green beards recognize and direct cooperation towards other greenbearded individuals. Consequently, a single locus controlling cooperative behaviour is referred to as a 'greenbeard'.

Because of its obvious fitness benefits, if a discrete greenbeard locus were to emerge it would be expected to rapidly sweep to fixation. Consequently, all individuals would display the greenbeard, leaving the signal devoid of information content[6–9], thus eliminating its role as an extant modulator of variation in cooperative behaviour and potentially rendering greenbeard genes relatively resistant to discovery[8]. Furthermore, although the discrete greenbeard scenario provides a compelling illustration of the concept, natural populations typically contain individuals that vary in cooperative traits, often with complex patterns of partner-specific behaviours. This phenomenon suggests that a simple discrete greenbeard system that simply distinguishes cooperators from non-cooperators would be insufficient to explain natural variation. Finally, in a real biological system, falsebeard cheating genotypes would be expected to emerge that display the greenbeard phenotype but do not cooperate[8,10–12]. Despite these failings, theory has demonstrated that all these problems can be mitigated if a greenbeard system is multi-allelic (or 'polychromatic' by analogy) because it can provide the specificity required for individuals to direct cooperation towards matching partners and stay ahead of the emergence of falsebearded individuals[13–15]. Such a polychromatic system would retain information content, with the evolutionary dynamics being akin to the evolution of genetic diversity in a host-pathogen system.

Because the requirements for a locus to act as a greenbeard are very restrictive[1], there have been relatively few reports of greenbeard genes. Examples of putative greenbeard loci that fulfil some criteria have, however, been described in diverse organisms. These loci regulate a broad range of recognition phenomena, thus hinting at their potential utility and evolutionary conservation. Putative helping greenbeards include *Dictyostelium discoideum csA*[16], *Neurospora crassa doc-1, doc-2, doc-3* (ref. 17); *Uta stansburiana OBY*[18], budding yeast *FLO1* (ref. 19), the *Botryllus schlosseri FuHC* locus[20], *Proteus mirabilis idsD* and *idsE*[21–23] and the vertebrate major

histocompatibility complex[24–26]. Putative harming greenbeards have also been described, including the fire ant *Gp9* locus[27] and loci controlling bacteriocin production and immunity in bacteria[28]. Interestingly, bacteriocins are highly polymorphic, with strains also often producing several different bacteriocins. Bacteriocins are thus able to convey highly specific and complex protection against other strains and could therefore be considered polychromatic, albeit regulating harming behaviour[8,14,28–30]. However, for most putative helping greenbeards, there is little evidence that any of the reported genes can explain complex natural variation in partner-specific patterns of engagement in cooperative behaviours as envisioned by Hamilton[1] and as often observed in nature. For example, even though sequence variation has been described for budding yeast *FLO1,* which modulates cell-cell adhesion and flocculation (a nominally cooperative trait)[19], this variation simply determines whether cells are competent to cooperate (that is, 'green' enough). Similarly, although sequence variation and IdsD/IdsE protein binding in *P. mirabilis*[21–23] can explain partner-specific interactions during swarming, the extent of allelic variation and thus capacity for complex partner-specific interactions is unknown. Despite this, both examples highlight the fact that cell adhesion proteins represent a standout candidate to encode a polychromatic greenbeard[31]. This is because they are localized at the cell surface, giving cells the ability to differentially adhere to other cells expressing the same molecule and thus can directly modulate cell behaviours[31]. Crucially, sequence variation could potentially generate a spectrum of beard colours, and thus provide the necessary specificity for identifying compatible partners.

The social amoeba, *D. discoideum*, provides a model to study the evolution and maintenance of social behaviour[32,33]. In response to starvation, several thousand individual *D. discoideum* amoebae aggregate together to undergo multicellular development, which ultimately results in the formation of a multicellular fruiting body composed of a stalk and sporehead. Multiple different genotypes will co-aggregate[34], making chimeric development an arena for cooperation and conflict. For example, conflict in chimeric development could arise over which cells contribute to the dead stalk versus the viable spores[33] and could result in deviations from clonal developmental strategies when in chimera[35,36]. Costs associated with conflict in chimeric development could be avoided by segregation away from socially incompatible individuals (thereby failing to engage in potentially cooperative interactions). Indeed, segregation has been reported between strains, with the degree of segregation shown to correlate to geographic and genetic distance[37]. These strains also exhibit high levels of polymorphism at two loci, *tgrB1* and *tgrC1*, which are thought to encode cell adhesion molecules[38–40]. Furthermore, elegant gene swapping experiments have shown that matching *tgrB1* and *tgrC1* alleles are required for co-aggregation[37,41,42]. These studies thus raised the possibility that a polychromatic greenbeard system based on TgrB1- and TgrC1-mediated cell adhesion could underlie cooperative behaviour in natural populations of *D. discoideum*.

Here we test this conjecture and demonstrate that a variable cell adhesion system represents a polychromatic greenbeard that underlies variation and co-existence in cooperative behaviour in a natural population of *D. discoideum*.

## Results

**TgrB1/TgrC1 sequence variation predicts segregation patterns.** To test whether the *tgr* genes could represent a polychromatic greenbeard, we first determined whether naturally co-occurring

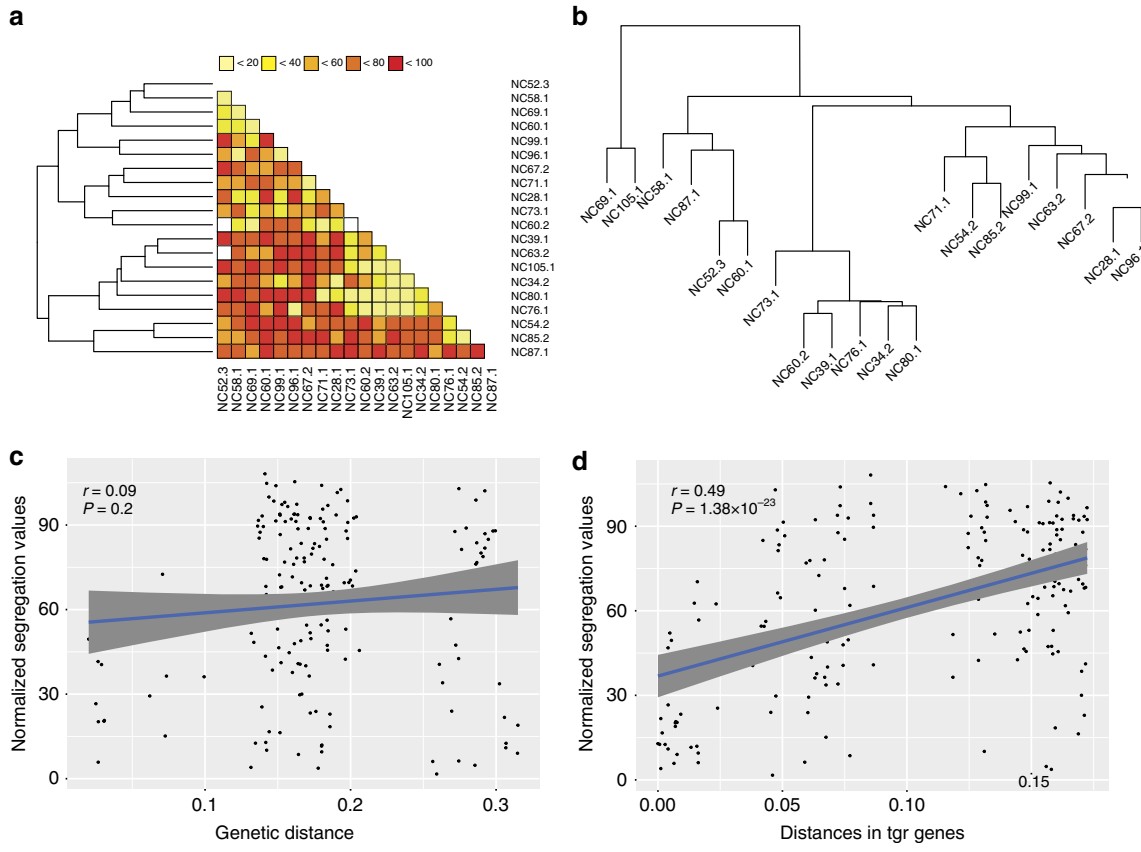

**Figure 1 | Patterns of segregation among 20 co-occurring *D. discoideum* strains.** (**a**) Clustering of pairwise segregation values. Pairwise segregation values were measured for 20 strains and clustered using Euclidean distances and a hierarchical clustering algorithm. Lighter coloured squares depict low segregation between strains, while darker colours depict high segregation. (**b**) Dendrogram of clustered Identity-By-State proportions among 20 co-occurring strains. Identity-By-State (IBS) proportions were computed from 30,444 filtered SNPs and clustered using complete hierarchical clustering. The associated dendrogram shows four clusters of varying sizes. (**c**) Relationship between genomic distances and segregation values. Pairwise genomic distances between strains were extracted from the dendrogram and plotted against normalized pairwise segregation values. The correlation between the two (visualized by the blue line, with the gray shading indicating the 95% confidence envelope) is not significant (Pearson's product moment correlation: $r = 0.09$ ; $P = 0.2$). (**d**). Relationship between tree distances inferred from concatenated TgRB1 and TgrC1 protein alignments and segregation values. Pairwise distances were extracted from the phylogenetic tree computed from concatenated TgrB1 and TgrC1 protein alignments. The correlation between tree distances and normalized pairwise segregation values (visualized by the blue line, with the gray shading indicating the 95% confidence envelope) is highly significant (Pearson's product moment correlation: $r = 0.49$; $P \ll 0.001$).

strains are able to 'choose' whom they partner with in a chimera. We measured the degree of segregation within pairwise mixtures of 20 strains isolated from the same North Carolina locale[34,35,43–45]. These strains were isolated from $1\,m^2$ patches of soil and have been shown to exhibit limited linkage disequilibrium, suggesting that recombination and mixing is common[33,44]. We found that the vast majority of pairwise chimeric mixes (167 of 207; 80.6 %) exhibited significant segregation compared to self-mix controls (Supplementary Fig. 1). Critically, we find that segregation is not a binary behaviour, but varies quantitatively from *ca.* 18 to 100% depending on the particular strains paired (Supplementary Fig. 1). Most importantly, strains exhibit considerable diversity in partner-specific segregation (Fig. 1a), with three-way mixes also following expectations based on segregation behaviour in pairwise interactions (Supplementary Fig. 2). Such partner-specific interactions result in non-transitive patterns, with hierarchical clustering based on the degree of segregation revealing little organization (Fig. 1a).

If a kin recognition process drives segregation, then the degree of segregation would be expected to reflect the overall genetic distance between strains. This is because common ancestry causes, on average, a similar degree of allele sharing across the whole genome. Therefore, a mechanism based on kin recognition would be expected to show a uniform relationship between allele sharing across the genome and segregation (that is, the average distance for the whole genome should be predictive of behaviour)[42]. To test this, we carried out whole genome sequencing of these strains. Hierarchical clustering of 30,444 single-nucleotide polymorphisms (SNPs) revealed a strong phylogenetic signal between the strains (Fig. 1b). However, overall genetic distance between strains does not predict the degree of segregation (Fig. 1c; Pearson's product moment correlation: $r = 0.09$, $P = 0.2$). We, therefore, tested whether a polychromatic greenbeard mechanism could instead drive segregation behaviour in *D. discoideum*. For a locus (composed of one or more genes) to underlie a greenbeard mechanism, we would expect it to fulfil two criteria. First, a greenbeard locus should exhibit a significant correlation between the allelic similarity of strains and their degree of segregation[8]. Second, a candidate polychromatic greenbeard locus must also harbour a level of functional variation that is sufficient to provide the necessary specificity to pairwise interactions underlying segregation. To test whether any genes

fulfilled these criteria, we firstly implemented a genome-wide association analysis to test whether the pairwise protein distances between strains (using a set of 6,532 genes with at least one non-synonymous polymorphism; Supplementary Fig. 3, Supplementary Fig. 4A and Supplementary Data 1) correlate with the pairwise segregation values. Considering the tests for the individual genes as being approximately independent, we set a conservative threshold based on a Bonferroni correction (using the Šidák equation) with a familywise error rate of 5%, which corresponds to a P value threshold of $7.85 \times 10^{-6}$. This analysis identified 86 greenbeard candidate genes that surpass this significance threshold, with the four largest correlations occurring for genes that map to the same local chromosomal region (Supplementary Fig. 4B and Supplementary Data 1). Second, we found that seven of the ten most polymorphic of these candidate genes (in terms of the number of alternative functional alleles, which in this case corresponds to nine or more alleles) all co-locate to that same genomic region, strongly implicating that region as containing a candidate greenbeard locus. Most strikingly, two genes within that region, *tgrB1* and *tgrC1*, stand out because they top the list of candidates based on each of the two criteria. They have the two highest levels of allelic diversity among the candidates (with 18 alleles each out of the 20 strains) and the highest levels of total sequence variation (synonymous and non-synonymous) in the genome (Supplementary Fig. 4C and Supplementary Data 1), while also showing the two largest correlations with segregation (see Supplementary Data 1 and Supplementary Fig. 5 for examples of segregation correlation for other highly polymorphic genes). Critically, although our analysis identifies two candidate genes, these show very tight physical linkage, as they share a 523 bp common promoter, and are biochemically coupled[37,38,41]. Hence, the pair can constitute a single locus (the '*tgr* locus'), as required by the greenbeard mechanism. Considering the entire functional *tgr* locus sequence, we find that there is sufficient allelic diversity for every strain to carry a unique allele, thus providing sufficient putative variation to explain the patterns of segregation observed. Further strong support for the *tgr* locus as a polychromatic greenbeard candidate comes from gene knockout and gene swapping experiments in isogenic laboratory strains, which have shown that a matching pair of *tgrB1* and *tgrC1* alleles is necessary and sufficient for attractive self-recognition and cell-cell adhesion[41].

To confirm the hypothesis that sequence polymorphisms at the *tgr* genes are consistent with expectations for a greenbeard locus, we next used Sanger sequencing to ensure that the polymorphisms were not derived from reads stemming from highly similar genes. Firstly, analyses of these final gene sequences revealed that, while the phylogenetic trees computed from the variants present at these genes are not identical (Supplementary Fig. 6), distances between strains of both *tgr* genes are very highly positively correlated (Pearson's product moment correlation: $r = 0.89$; $P = 1.84 \times 10^{-128}$; Supplementary Fig. 7A), supporting the idea that the genes show concerted evolution as a single functional locus. Differences in the specific topology of the phylogenetic trees for the two genes only occur where branch lengths are very short and bootstrap values are very small (Supplementary Fig. 6, haplotype B), suggesting that there is not enough variability between the sequences of these strains to infer a reliable phylogeny. However, the overall topologies of the trees are mostly identical (Robinson Foulds distance of 14). Importantly, pairwise distances derived from a maximum likelihood tree for the *tgr* locus, which was computed using the concatenated alignments, were strongly correlated with the segregation patterns (Pearson's product moment correlation: $r = 0.49$, $P = 1.38 \times 10^{-23}$; Fig. 1d, Supplementary Fig. 7B,C for

separate correlations for the two genes). Furthermore, because genetic distances at the *tgr* genes are only weakly correlated with the genomic distances between strains (Pearson's product moment correlations: *tgrB1*: $r = 0.24$; $P = 0.001$, *tgrC1*: $r = 0.21$; $P = 0.004$; Supplementary Fig. 7D,E), but are far more strongly correlated with the degree of segregation, this supports the conclusion that *tgr* gene evolution is occurring more rapidly than the rest of the genome (that is, the *tgr* genes are diverging faster than the background rate). Finally, analyses of SNP patterns revealed two haplotype groups (A and B) at the *tgr* locus. These groups are also seen when strains from other geographic regions are added to the alignment, suggesting that these groups are evolutionarily ancient (Supplementary Fig. 8A). Analyses of SNP patterns did also reveal substantial amino acid variation at a large number of positions both within and between haplotype groups, with several different alternative amino acids often seen at a single position (Supplementary Fig. 8B). Most of the variable positions (238 out of 295) exhibited a Shannon entropy value of $< 1$ (ref. 46), indicating conserved sites experiencing purifying selection[47]. Of these, positions with Shannon entropy values close to 1, correspond to polymorphisms that separate the A and B haplotype groups. Critically, there are also 57 highly variable sites clustered at the N-termini of the proteins that show Shannon entropy values of $> 1$ suggesting diversifying selection (Supplementary Fig. 8B). A fixed effect likelihood method supported this inference. Taken together, these results indicate that diversifying selection within the N-termini of TgrB1 and TgrC1 proteins rather than genome-wide ancestral relatedness is the primary driver of cooperative aggregation in co-existing *D. discoideum* populations.

**TgrB1–TgrC1 interaction strength determines segregation.** Homophilic cell adhesion molecules could act as greenbeards if their extracellular domain allows matching copies to be recognized on other cells whilst intracellular domains confer subsequent cellular responses[31]. Consistent with this idea, Tgr proteins have been shown to play roles in cell–cell adhesion and cell signalling[38–40]. However, it is currently unknown how sequence differences cause variation in 'recognition'. One simple explanation is that TgrB1/TgrC1 sequence variants could confer specificity to the interaction (allowing for non-linear and potentially non-transitive patterns) and result in categorical differences in binding. We, therefore, tested whether measures of *in vitro* TgrB1/TgrC1 binding could predict the segregation patterns of four representative strains (three from haplotype group A and one from group B), and whether binding was a simple function of sequence divergence. Group A strains exhibit greatest segregation with the most genetically distant group B strain, NC34.2 (Fig. 2a). However, in some cases segregation between strains within group A (for example, NC71.1 versus NC96.1: 41% or NC96.1 versus NC52.3: 50%) can be almost as large. Importantly, the chosen strains highlight the non-transitive nature of segregation patterns. Although a hierarchy of segregation can be seen against strain NC34.2, with NC34.2 < NC71.1 < NC52.3 < NC96.1, a different hierarchy is observed when segregation is measured against strain NC96.1, with NC96.1 < NC71.1 < NC52.3 < NC34.2 (Fig. 2a).

We next tested whether Tgr protein interaction strength could predict segregation behaviour. Indeed, previous studies have shown that Tgr proteins mediate cell-cell interactions[39,40], while TgrB1 and TgrC1 interactions are required for clustering and adhesion complex formation[41]. Moreover, specific regions have been defined that are required for *in vitro* protein interactions when isolated from *D. discoideum* extracts[38].

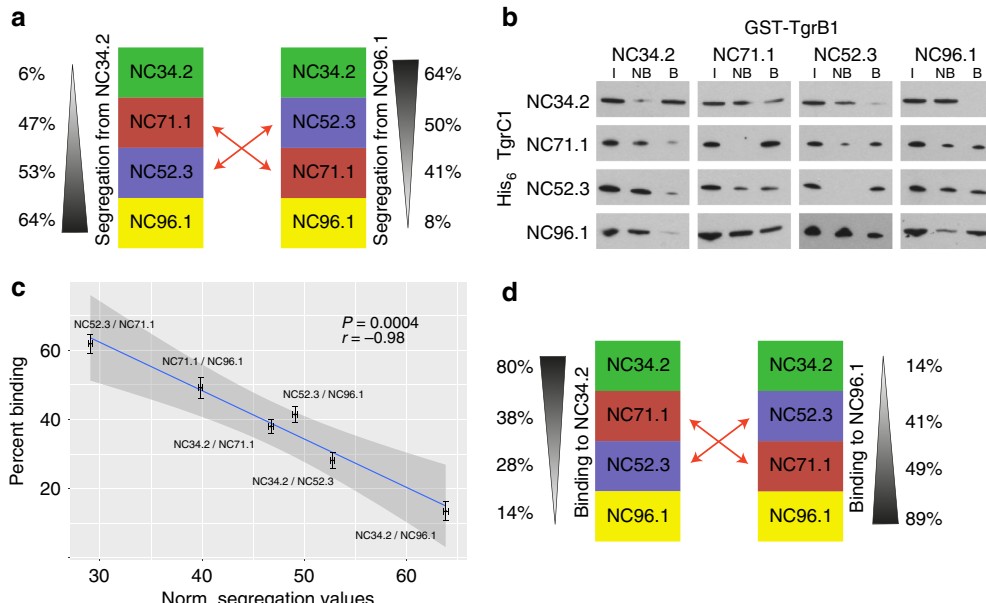

**Figure 2 | Segregation between strains can be predicted by TgrB1/TgrC1 binding.** (**a**) Strains exhibit non-transitive segregation behaviour. Segregation values between the four strains are not transitive. Numbers on the left show segregation values of mixes containing NC34.2 ordered from top to bottom and numbers on the right show segregation values of mixes containing NC96.1 ordered from bottom to top. Black triangles depict the degree of segregation. Red arrows depict the non-transitive swap of strains NC52.3 and NC71.1. Mean segregation values of all pairings within each hierarchy are significantly different (one-way analysis of variance (ANOVA); $P < 2.2 \times 10^{-16}$). (**b**) Immunoprecipitation of bacterially expressed $His_6$–TgrC1 and GST–TgrB1 from four different strains. The highest binding was observed between proteins from the same strain. The weakest binding was observed between Tgr proteins from NC34.2 and NC96.1. Varying levels of protein binding were observed between the Tgr proteins from other strains (I = input protein, NB = not bound protein and B = bound protein). (**c**) Scatterplot of binding and segregation values. For each pairwise strain comparison, normalized segregation values were plotted against percent binding. A minimum of six biological replicates were performed for each protein pairing. Y axis error bars depict s.e. of the mean percentage of bound protein in each fraction of the biological replicates, quantified by measuring band intensities on western blots using ImageJ[80]. X axis error bars depict s.e. of the mean normalized segregation value derived from 1000 bootstrapped samples (see methods). Blue line depicts highly negative and significant (Pearson's product moment correlation: $r = -0.98$, $P = 0.0005$) correlation between segregation values and protein binding for mixes of four chosen strains (gray shading: the 99% confidence envelope). Self-mixes are excluded. (**d**) Strength of binding predicts non-transitive segregation behaviour. Binding is not transitive. Numbers show binding of the Tgr proteins. Black triangles depict the strength of binding. Red arrows depict the non-transitive swap of strains NC52.3 and NC71.1. This pattern mirrors the non-transitive behaviour of the segregation values (shown in panel **a**). Mean binding values between each pairing in the hierarchy are significantly different (one-way ANOVA; $P < 2.2 \times 10^{-16}$ using adjusted $P$ value 0.05).

However, because the vast majority of highly variable sites do not occur in regions previously described as being required for protein–protein interaction (Supplementary Figs 8-10), we expressed almost full-length TgrB1 and TgrC1 proteins from these strains in bacteria (amino acids 65–861 and 57–867 respectively or 88.47% and 98.5% of each coding sequence, see Methods section for details). Each protein thus contained the domains previously described to mediate protein interactions, as well as the highly polymorphic N-terminal region of unknown function. In fact, the sequence within the known binding domain and C-terminal regions is actually identical for TgrB1 in three of the chosen strains (Supplementary Figs 9 and 10 for full alignments and primer positions). The strength of protein interactions was tested in pairwise co-immunoprecipitation experiments (Fig. 2b). In all cases, the strength of binding (Fig. 2d) strongly correlates with the degree of segregation (Fig. 2c) and even predicts the non-transitive swap (Fig. 2d). It is important to note that binding patterns were reproducible over a 125-fold range of protein concentrations (Fig. 2c, Supplementary Figs 11 and 13). Therefore, even though Tgr protein concentrations on the cell surface are unknown, binding patterns are extremely robust. Furthermore, the observed binding patterns necessarily imply that the degree of binding cannot simply be a direct function of the number of SNPs, since such a linear relationship would not allow for the non-transitive swapping in relative binding. Finally, since the TgrB1 sequence within the known binding domain and C-terminal regions is identical in three of the tested strains (Supplementary Fig. 9), taken together, these results thus strongly suggest that precise combinations of Tgr protein polymorphism within the N terminus can explain diverse partner-specific recognition driven by protein binding strength, while providing the specificity required for the locus to act as a polychromatic greenbeard system.

**Greenbeard recognition protects against costs of chimerism.** To act as a greenbeard, *tgr* gene dependent recognition must ultimately elicit a response that favours individuals to preferentially cooperate with individuals with whom they share a compatible greenbeard variant. In *D. discoideum*, chimeric aggregation can result in conflicts that result in fitness costs manifested during the slug or fruiting body stage. For example, chimeric development may cause alterations in developmental patterning, coordination or timing and has been shown to affect the migratory behaviour of slugs[48,49]. We, therefore, tested whether a polychromatic greenbeard system could mitigate this known cost of chimerism. For this,

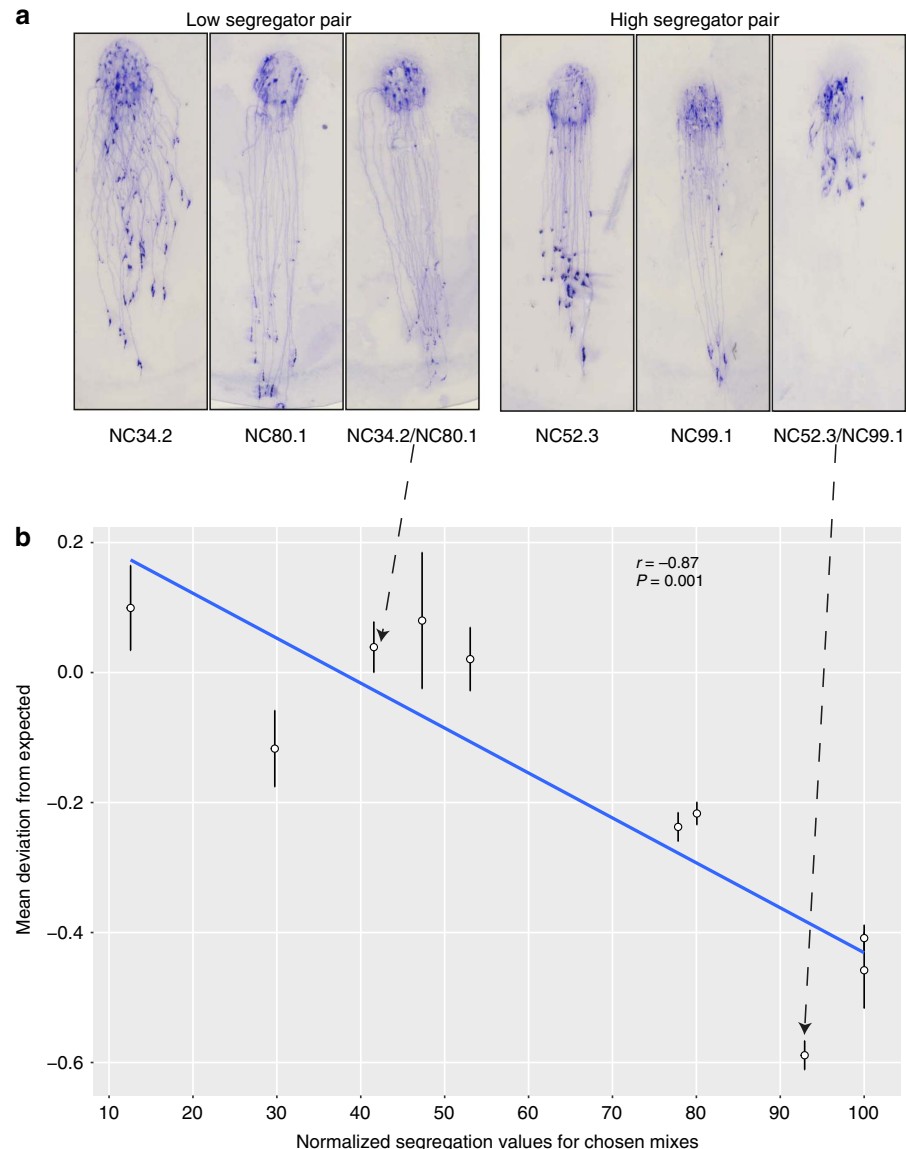

**Figure 3 | Strains with higher segregation values exhibit reduced slug migration.** (**a**) Representative slug trail images from clonal and chimeric pairs showing low (41%) and high segregation (93%). The circular area at the top represents the spot where cells were initially plated and is the point of origin of the slug trails. Slugs were allowed to form and migrate toward a unilateral light source at the bottom of the dish over a period of 30 h before the trails were transferred from the surface of the agar onto a PVC disc and then stained with Coomassie Brilliant Blue. (**b**) Slug migration distances across ten different pairwise strain combinations. The mean migration distances were calculated for each clone and chimeric mix from four replicates (error bars depict s.e. of these four replicates). Expected chimeric values for each pair were then calculated as an average of the slug migration of each clone in the mix. Slug migration values are expressed as a deviation from these expected values. Mean slug migration distances of the high segregating pairs (>90%) are significantly shorter (Pearson's product moment correlation; $r = 0.87$, $P = 0.001$) than the expected migration distances based on the slug migration of the two respective clones in isolation.

we exploited the fact that segregation is mostly eliminated when strains undergo development on a non-natural agar substrate (Supplementary Fig. 12). The reason for this is unknown, but development on soil is thought to result in a more physiologically relevant and challenging physical environment for cell behaviours such as cell migration, which is important during aggregation. Indeed, soil development has previously been shown to uncover phenotypic defects caused by null mutations that are silent on agar development[16,50]. Hence, we were able to measure the consequences of chimeric development for pairs of strains that would naturally avoid chimerism through segregation. We compared slug migration efficiency during clonal and chimeric development using

five strain pairings that show little or no segregation on soil or agar and five pairings that exhibit high levels of segregation on soil, but little segregation on agar (that is, forced chimerism). Strains with the highest segregation exhibited much reduced slug migration when forced into chimeric development (Fig. 3; Student's t-test: $r = -0.87$, $P = 0.001$), demonstrating that there is a cost to chimerism manifested in motility at the slug stage, but only when strains are forced to aggregate with strains they would otherwise segregate from. Thus, these results are consistent with the idea that the variation at the Tgr locus evolved as a greenbeard based mechanism to reduce the fitness costs caused by chimeric shifts in behaviour affecting slug motility.

## Discussion

Taken together, these experiments demonstrate that the *Dictyostelium* Tgr locus exhibits all of the properties required for it to act as a polychromatic greenbeard—it encodes the tag, determines recognition, and modulates engagement in cooperative behaviour. Most critically, we show that sequence variation at the Tgr locus generates a homophilic binding spectrum that allows individuals to identify appropriate partners with whom to engage in cooperation. By providing this specificity, the locus stabilizes cooperation in the face of the selective pressure for the emergence of falsebearded cheaters by providing information that can be used to differentiate compatible from incompatible partners.

## Methods

**Strains.** All strains were isolated from soil samples collected in Little Butt's Gap, NC, USA (coordinates: 35°46.317' N; 82°20.533' W)[44] and obtained from the Dicty Stock Center [51]. The following strains were used for the segregation analysis: NC105.1, NC28.1, NC34.2, NC39.1, NC52.3, NC54.2, NC58.1, NC60.1, NC60.2, NC63.2, NC67.2, NC69.1, NC71.1, NC73.1, NC76.1, NC80.1, NC85.2, NC87.1, NC96.1, NC99.1.

**Measuring segregation.** Each strain was grown to mid-exponential phase on SM-agar plates in association with *Klebsiella aerogenes*. Cells were harvested from plates and washed three times in KK2 buffer (14 mM K2HPO4 and 3.4 mM K2HPO4, pH 6.4) before resuspending at a density of $10^7$ cells per ml in KK2. To label cells, 50 μM of CellTracker Green CMFDA was added to the cell suspension (or 50 μM DMSO to unlabelled cells) and incubated at 22 °C with shaking for 30 min. Cells were then washed twice in KK2 before a further incubation with shaking in KK2 for 30 min. Cells were then resuspended at a density of $10^8$ cells per ml in KK2 before mixing in equal proportions with unlabelled cells in either pairwise mixes (50:50) or three-way mixes (33:33:33) For the segregation measures 10 μl of cell mixture was then deposited in wells of a 24 well dish containing ~1.25 g washed sharp horticultural sand (Keith Singleton) and 250 μl KK2. Dishes were incubated at 22 °C in a humid box until fruiting body formation (a minimum of 24 h). Spores were harvested from individual fruiting bodies in spore buffer (KK2 containing 20 mM EDTA and 0.05 % NP40) and the proportion of fluorescent and non-fluorescent spores in each fruiting body was analysed using a CYAn flow cytometer. In total, 207 pairwise mixes and 2 three-way mixes were conducted and percentages of different genotypes per fruiting body (FB) recorded for an average of 8.4 FB per mix using a total of 1,743 FBs in pairwise mixes and an average of 109.5 FBs per mix using a total of 219 FBs in three-way mixes. Segregation for each combination of strains (*i* and *j*) was computed as the s.d. of the percentages of a given strain in different FBs of this mix ($std_{i,j}$). Because the overall relative representation of a strain across all FBs of a particular mix sets an upper limit to the measure of segregation, we calculated normalized segregation values ($nsv_{i,j}$) by dividing the raw estimate of segregation ($std_{i,j}$) by its maximum possible value:

$$nsv_{i,j} = 100 \times \left( \frac{std_{i,j}}{\sqrt{mean_{i,j} \times (1 - mean_{i,j})}} \right) \quad (1)$$

These normalized values can therefore be interpreted as percent segregation. To infer a cutoff that distinguishes segregation from non-segregation, we estimated the null distribution of $nsv_{i,j}$ by fitting a gamma distribution to the $nsv_{i,j}$ values from self-mixes (that is, clones) using the R packages MASS and lmom (shape = 4.26, rate = 0.51) and setting the threshold as the 99% percentile (which corresponds to a $nsv_{i,j} < 18$ being non-segregation and $nsv_{i,j} \geq 18$ indicating segregation).

**DNA extraction.** $10^9$ *Dictyostelium* cells of each strain were harvested after growth on SM-agar plates in association with *K. aerogenes*. Amoebae were separated from bacteria by differential centrifugation at 1800 rpm for 2 min and multiple washes in KK2 buffer before resuspending in 10 ml nuclei buffer (40 mM Tris pH 7.8, 1.5% sucrose, 0.1 mM EDTA, 6 mM MgCl2, 40 mM KCL, 5 mM DTT, 0.4% NP40) and vortexing at 8,000 r.p.m. for 10 min. The pellet was then resuspended in EDTA (final volume 100 μl, final concentration 100 mM EDTA) before adding 250 μl 10% sodium lauryl sarcosyl and incubation for 20 min at 55 °C. After this, 250 μl 4M ammonium acetate was added and centrifuged at 1,800 r.p.m. for 15 min. The supernatant was transferred to a clean tube before adding 2 volumes of ethanol and vortexing at 8,000 r.p.m. for another 10 min. The pellet was then washed in 70% ethanol, drained, and resuspended in 10 μl TE. Genomic DNA was treated with epicentre RiboShredder RNase blend followed by Phenol-chloroform extraction and ethanol precipitation. The final pellet was dissolved in TE buffer.

**Sequencing.** Genomic libraries from all strains were prepared using the Illumina TruSeq kit and sequenced using 100 bp paired-end reads (150 bp insert) on an Illumina Hiseq 2500. In addition, mate-pair libraries (3 kbp insert) were sequenced from strains NC32.2, NC80.1, NC85.2, NC76.1, NC52.3.

**Mapping.** Paired-end reads from all samples were quality checked and filtered using the IlluQC_PRLL.pl script (v2.3) from the NGS QC toolkit[52] and a read length cutoff of 70 % together with a quality score cutoff of 20. In all samples more than 90 % of the reads were retained, which were then mapped against the *Dictyostelium discoideum* reference genome sequence version from 30 April 2012 (ref. 53), (masking the inverted repeat on chromosome 2) using bowtie2 version 2.0.0-beta5 (ref. 54) and very-sensitive end-to-end mapping parameters requesting the best out of ten alignments. Overall alignment rate varied between 35 and 98%. An analysis of the samples with low alignment rates revealed a significant contamination with bacterial sequences. Reads from the mate-pair libraries were adapter trimmed in two steps using cutadapt version 1.3 (ref. 55). In a first step the Illumina Truseq adapter was trimmed using an error rate of 0.1 and the ends of the reads were quality trimmed with a quality value of 20. The second step removed the Illumina mate pair adapter with an error rate of 0.1. Reads were then mapped with the same parameters as the paired-end reads and samfiles from the same strains combined using SAMtools (ref. 56), after which mitochondrial reads were separated to ensure a more even read coverage.

**SNP calling and clustering.** SAMfiles of all samples were further processed using picard tools version 1.106 (http://picard.sourceforge.net). SAMfiles were sorted (SortSam.jar), duplicates marked (MarkDuplicates.jar) before read groups were added (AddOrReplaceReadGroups.jar) and BAM indices build (BuildBamIndex.jar). The GATK[57–59] was then applied to the resulting BAMfiles to realign indels and remove duplicates. SNPs were called for all samples combined using the UnifiedGenotyper with parameters -ploidy 1, -glm SNP and a minimum phred-scaled confidence threshold of 100 to call and emit variants. Called variants were hard filtered using filters 'QD < 2.0', 'FS > 60.0' and 'MQ < 30.0'. Variants passing the filter were annotated using SNPeff[60].

Filtered SNPs were LD pruned (LD threshold: 1.0) and Identity-By-State (IBS) proportions computed from remaining 30,444 SNPs using the Bioconductor package SNPrelate and R[61,62]. These were then clustered using complete hierarchical clustering.

**Assembly of nucleotide sequences.** To obtain sequences of genes showing a high number of filtered SNPs, trimmed reads were assembled using velvet Version 1.2.10 (refs 63–65) and k-mer values between 21 and 85. Contigs from all assemblies were then searched against a database of 13,409 known *D. discoideum* transcripts downloaded from dictyBase[66] at 02.05.2013 using BLAST version 2.2.27 (parameters: blastn, *E* value $\leq 10^{-5}$). All sequences with either tgrB1 or tgrC1 as best hit were further assembled using CAP3 (ref. 67) and then manually corrected using CLC Genomics workbench 6.

**Sanger sequencing of tgr genes.** SNPs in highly polymorphic regions of tgrB1 and tgrC1 were further validated by Sanger sequencing (tgrC1 forward primer: 5′-GAACCCAGAACTGAAATGGCAC-3′; tgrC1 reverse primer: 5′-GTAATAGGCAAGAGCACC-3′; tgrB1 forward: 5′-CAATATTAGT AGTAGTGGGATTC-3′; tgrB1 reverse: 5′-CCGAAACCAGGTCCTAGAAC-3′). Primer locations are highlighted in Supplementary Figs 9 and 10. All SNPs called from the mapping data in these regions were validated in the Sanger sequences.

**Alignments and Phylogenetic trees of *tgrB1* and *tgrC1*.** Assembled nucleotide sequences were first translated into protein sequences using CLC Genomics workbench 6 and aligned using ClustalW2 (http://www.ebi.ac.uk/Tools/msa/clustalw2/). After removal of gapped sites, phylogenetic trees were constructed for each protein separately and the concatenated alignment using phyml, the LG model, estimated amino acid frequencies, best of NNI and SPR search, estimated proportion of invariable sites, sites, and optimized tree topology, branch lengths, and rate parameters[68]. Additional full length tgrB1 and tgrC1 nucleotide sequences of strains QS9, QS14, QS17, QS23, QS34, QS36, QS37, QS38, QS40, QS41 and QS47, which were sequenced in previous studies were downloaded from NCBI, translated, and a new concatenated alignment and tree computed.

**Alignments and phylogenetic trees of other genes.** Sequences for all genes of all strains were generated using the AX4 reference sequence and the filtered vcf file using vcf-consensus from the vcftools package[69]. Sequences from different strains for each gene were then combined using a custom perl script and aligned using ClustalW. Shannon entropy and the number of synonymous and non-synonymous SNPs for each alignment were extracted using the R packages bio3d[46,70,71] and SeqinR[72].

**Statistical analyses and tests for selection.** All correlations, linear models, figures, and data manipulations were conducted using R version 3.2.0, Rstudio version 0.98.1091 and R packages dplyr[73], tidyr[74], MASS[75], lmom[76], ggplot2 (ref. 77), gplots[78].

To identify sites under diversifying selection nucleotide alignments were analysed using the FEL method on the datamonkey.org public server. Shannon entropies for each site were computed using the R package bio3d[46,70,71].

To determine whether the segregation values differ significantly for the 6 mixes between strains NC34.2, NC52.3, NC71.1 and NC96.1, 10 out 40 values of the percentages of the single fruiting body were randomly sampled 1,000 times and the normalized segregation value computed for each sample. These were then used to conduct 15 pairwise $t$-tests and a Šidák corrected alpha of 0.0034.

**In vitro expression of TgrB1 and TgrC1.** For GST-TgrB1 protein expression, a fragment of TgrB1 from each isolate was amplified by PCR (forward primer: 5′-cgcGTCGACAATTTCCTTACAAGAATCTG-3′, reverse primer: 5′-cgcGC GGCCGCAAGGCGATTTCAGTAGC-3′) and subcloned into the pGEX vector between the SalI and NotI sites. For His$_6$–TgrC1 protein expression, a fragment of TgrC1 from each isolate was amplified by PCR (forward primer: 5′-cgcGTCGAC AAGAACCCAGAACTGAAATGGCAC-3′, reverse primer 5′-cgcGCGGC CGCGTAATAGGCAAGAGCACC-3′) and subcloned into the pET22b+ vector between the SalI and NotI sites. For large scale purification of GST-TgrB1 proteins, transfected JM101 E. coli cells were cultured overnight in LB broth at 37 °C in the presence of 50 μg ml$^{-1}$ ampicillin. A diluted culture was then grown at 30 °C until a density of A$_{600}$ 0.6–0.7 was achieved (∼1 h 30 min). Protein expression was induced by addition of 0.5 mM isopropyl thiogalactoside for 4 h. Cells were harvested and the cell pellet was resuspended in sonication buffer (0.5 M NaCl, 10 % glycerol, 1 mM EDTA, 0.1 mM phenylmethyl sulphonyl fluoride and 1 mM dimethylsulphoxide in PBS). Cells were lysed by sonification and centrifuged at 10,000 r.p.m. for 15 min. 1 % Triton X-100 was added to the supernatant before protein purification by glutathione-Sepharose affinity chromatography. For large scale purification of His$_6$–TgrC1 proteins, transfected BL21(DE3) E. coli cells were cultured overnight in 2x TY broth (1.6 % w/v Tryptone, 1 % w/v yeast extract and 0.5 % w/v NaCl) containing 50 μg ml$^{-1}$ ampicillin. A diluted culture was then grown at 30 °C until a density of A$_{600}$ 0.6–0.7 was achieved (∼1 h 30 min). Protein expression was induced by addition of 0.5 mM isopropyl thiogalactoside for 4 h. Cells were collected and then lysed in lysis buffer (2% SDS, 6 M urea, 10 mM Tris, 10 mM imidazole, pH8). Lysed cells were centrifuged at 10,000 r.p.m. for 15 min before protein purification on an Ni$^{2+}$-nitrilotriacetic acid agarose column[79].

Antibodies used for these experiments were ordered from Abcam (www.abcam.com): antiGST (catalogue number: ab9085; used 1:10,000) and Goat antirabbit IgG (horse radish peroxidase) (catalogue number: ab205718; used 1:10,000).

**Pull down assays.** To analyse the binding affinity of TgrB1 and TgrC1 from different isolates, 25, 5, 0.5 or 0.2 μg of GST–TgrB1 was incubated with equal amounts of His$_6$–TgrC1 from either the same isolate in control experiments, or a different isolate, for 1 h at 4 °C in 0.5 ml binding buffer (50 mM NaH$_2$PO$_4$, 0.5 M NaCl (pH 8.0)). His$_6$–TgrC1 complexes were isolated on a Ni$^{2+}$-nitrilotriacetic acid agarose column. Bound and unbound proteins were subjected to SDS/PAGE before immunoblotting with a horse radish peroxidase-conjugated anti GST antibody to detect GST–TgrB1. Western blots were imaged using a densitometer (Supplementary Fig. 13) and the amount of protein in each fraction was determined by measuring band intensity using ImageJ.

**Slug migration assays.** Each strain was grown to mid-exponential phase on SM-agar plates in association with K. aerogenes. Cells were harvested from plates and washed three times in KK2 buffer (14 mM K2HPO4 and 3.4 mM K2HPO4, pH 6.4) before resuspending at a density of $5 \times 10^7$ cells per ml in saline buffer (10 mM NaCl, 10 mM KCl, 3 mM CaCl$_2$). A 20 μl spot of cells was plated onto the middle of a 10cm charcoal water agar plate (0.5% activated charcoal, 1.5% Phytagel (Sigma Aldrich, Inc.) agar). The plates were incubated at 22 °C for 30 h in darkened boxes with a pinhole at one end as a lateral light source. Slug trails were transferred to PVC discs, stained with 0.6% Coomassie Briliant Blue R (Sigma Aldrich, Inc.) in ethanol:acetic acid: water (5:1:4) for 5 min with shaking, and destained in 10% acetic acid (Sigma Aldrich, Inc.). Slug trail lengths were measured by tracing scanned trails using ImageJ software (version 2.0.0 (ref. 80). A mean value for slug trail length was then calculated for each clone and pairwise mix in each experiment. The expected migration distance for pairwise mixes was calculated as an average of the migration distances of the two clones in isolation from the same experiment. Slug migration values for each pairwise mix were then expressed as a deviation from this expected value. In total, four experiments were performed.

**Code availability.** All scripts used to gather and analyse the data are available on request.

**Data availability.** Raw sequencing reads were uploaded to the NCBI Sequence Read Archive (SRA) with the project identifier SRP071575 (http://www.ncbi.nlm.nih.gov/Traces/sra/sra.cgi?study=SRP071575). All raw data

(measurements for single fruiting bodies on soil and agar, images and quantifications of westerns, slug migration images) are available on request.

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

## Acknowledgements

We thank David Haig, Phil Madgwick, Laurie Belcher, Joan Strassmann and David Queller for useful discussions about polychromatic greenbeard gene properties. This work was supported by funding from the Biotechnology and Biological Research Council and Natural Environment Research Council to C.R.L.T. and J.B.W. (NE/H020322/1; BB/M007146/1); and a Wellcome Trust Investigator Award to C.R.L.T (WT095643AIA).

## Author contributions

N.G. performed the bioinformatics analyses of the sequencing data. N.G. and J.B.W. performed the statistical analyses. J.A.H. and K.P. performed the segregation measurements. K.P. performed the immunoprecipitation assays. B.S. performed the slug migration assays. C.R.L.T. and J.B.W. conceived the study, supervised the project and wrote the manuscript. All authors discussed the results and contributed to writing or commenting on the manuscript.

## Additional information

**Competing financial interests:** The authors declare no competing financial interests.

