## [Peer Review File · Nature Communications]

Reviewers' comments:

Reviewer #1 (Remarks to the Author):

This study reports the characterisation of a polychromatic greenbeard gene, involved in cell-adhesion-based cooperation in a social amoeba. During starvation, the cells in a local neighbourhood aggregate together, and form multicellular 'slugs' that move elsewhere and form fruiting bodies. The researchers investigate assortment of genotypes in chimaeric aggregates, finding that assortment is strongly affected by allelic variation segregating at the *tgr* locus and not by whole-genome genetic similarity, suggesting a greenbeard effect rather than genetic kin discrimination. They show that the assortment is modulated by interactions between this locus's gene products, and that there is a benefit to assortment owing to chimaeric slugs having reduced motility.

I can't comment in detail on the experimental methods, except to say they seem appropriate. Instead, my focus is on the theoretical motivation for the study and the conceptualization of the results.

1. I agree with how the authors have interpreted their findings, i.e. that this is a polychromatic greenbeard. However, I think there needs to be a bit more care with regard to the claims for novelty and setting the study in its proper context within the existing literature. In particular, bacteriocins have been conceptualised in terms of the greenbeard effect - e.g. by Gardner et al (2004, Proc B 271: 1529), Gardner & West (2010, reference 12 of this ms), West & Gardner (2010, reference 13 of this ms) and Inglis et al (2011, Am Nat 178: 276) - and models of the evolutionary maintenance of bacteriocin diversity have a 'chromodynamics' flavour - e.g. see Biernaskie et al (2013, J Evol Biol 26: 2081), and references therein - even if this particular term hasn't always been employed. The bacteriocin scenario is different from the present cell-adhesion scenario in that it is harming rather than helping, so perhaps a distinction could be made on that basis, but this would need to be spelled out.

2. Page 2, "... an abstract scenario in which... a metaphorical green beard". In Dawkins's (1976) scenario, it's a literal green beard, not a metaphorical one.

3. Page 2, "... eliminating its role as an extant modulator of variation in cooperative behaviour" - Gardner & West (2010) suggested that this may render greenbeards relatively resistant to discovery.

4. Page 2, the term "falsebeard" was coined by Gardner & West (2010).

5. Page 4, middle paragraph, it may help the reader to point out explicitly that the greenbeard effect is conceptually distinct from genetic kin discrimination, as these two have often been confused and conflated in the literature. Dawkins (1982, The extended phenotype) and Gardner & West (2010, especially figure 1) spell out the difference.

As is my policy, I waive anonymity

Andy Gardner

Reviewer #2 (Remarks to the Author):

In the manuscript submission "A polychromatic 'greenbeard' locus determines patterns of cooperation in a social amoeba", Gruenheit et al. provide evidence that polymorphisms at previously characterized self-recognition complex Tgr contribute to variations in cooperation observed in wild isolates of the social amoeba, *Dictyostelium discoideum*. The authors pair 20 wild isolates of *D. discoideum* to show that there are different degrees of separation between different wild isolates during fruiting body formation. This effect is caused by polymorphisms at a single genomic locus (Tgr) rather than overall genetic differences. Polymorphisms at the Tgr locus result in partner specific protein-protein binding strength differences (as shown by co-immunoprecipitation assays), that potentially mediate recognition specificity. The authors further show, via mixing of amoeba from different wild isolates, that a reduction in cooperation between incompatible partners, decreases the overall fitness, as assessed by slug migration.

The results of this paper are informative and well presented, yet not completely new. The Tgr locus has previously been shown to be polymorphic and necessary and sufficient for self-/non-self recognition during fruiting body formation of *D. discoideum*. This manuscript provides an extension to the initial finding showing that the polymorphisms of the Tgr locus mediate a 'polychromatic' effect. However, the significance of the study paper would be substantially increased by adding additional biochemical data on the hypothesis that the strength of protein binding is the underlying cause of the "polychromatic" greenbeard effect. The data as they are currently presented is preliminary. An example of assessing physical interaction of adhesion proteins has been biochemically characterized by the Gibbs laboratory (see below). Data supporting the hypothesis that binding activity of the Tgr proteins is the biochemical basis of the "polychromatic" aspect of these proteins is not fully substantiated by the data presented in this paper.

The introduction to the paper describes the greenbeard effect and introduces the theory underlying this phenomenon. However, the authors should cite some additional literature outside of the *D. discoideum* field. They claim that there have been relatively few reports of potential greenbeard genes, although reports of greenbeard genes have existed for at least 15 years (see Keller and Ross (1998) *Nature*; as an example). In a recent study of the filamentous fungus *Neurospora crassa*, a 'polychromatic' greenbeard example has been characterized in detail (Heller et al. (2016) *PLoS Biology*). Other examples for polymorphic genes that mediate non-self recognition and could be considered greenbeard genes are described in *B. schlosseri* (see work of the De Tomaso group) and *Proteus mirabilis* (see work of the Gibbs group). These studies should be included in the introduction in addition to the budding yeast *FLO1* because they show that the greenbeard effect is found in a wide group of organisms and therefore is of interest for a broader audience.

Additional information on the model for Tgr function and past biochemical characterization should be explicitly described in the paper.

The results and discussion section needs some revision:

The authors want to test if strains are able to choose their partners in chimeras (p. 3), but then only test segregation in pairwise mixtures. It would be interesting to see if segregation can be confirmed if more than two strains are mixed. For example, in a mixture of three strains (two with strong affinity, one with weak affinity), there should be a clear difference in segregation. The two strains with strong affinity should not segregate and appear in the same FBs while the strain with weak affinity to the other strains should segregate.

The authors sequence 20 wild isolates of *D. discoideum* and map the sequence back to the reference strain. Is it possible that additional loci show significant polymorphism between the strains and the reference sequence due to an inability to map back to the genome sequence of the reference strain? Was de novo assembly of any of the 20 genomes performed to test this hypothesis? Many microbes contain a number of loci that are highly polymorphic and that show diversifying selection. Is the Tgr locus the only identified locus? The authors claim that 'for the vast majority of these genes, the genetic distance between strains mirrors the pattern of overall distance', (p. 4). Does vast majority of genes mean that there are other loci beside the tgr locus that show a larger correlation with segregation behavior than the genome-wide pattern? If so, the authors should mention these loci and explain why they focused on the Tgr locus. In addition, a plot comparing the relationship between tree distance and segregation values for several of the identified polymorphic loci could be included as a control.

'...supporting the idea that the genes show concerted evolution as a single functional locus' (p. 4). A comparative phylogenetic analysis for each gene would help to visualize that the tree structure is indeed the same.

Co-immunoprecipitation experiments: In the materials and methods, it is indicated that "a fragment of TgrB1 from each isolate was amplified by PCR....". What fragment? How was this region chosen? This information is not sufficient to evaluate whether the methods and results are accurate for the co-immunoprecipitation experiments. Additionally, this section of the paper should be expanded with additional experiments to support the contention that variation in binding of the Tgr protein is the biochemical basis of the "polychromatic" aspect of these proteins. The authors should look at the paper from the Gibbs lab on the characterization of adhesion proteins that mediate self/nonself interaction in *Proteus* for examples (Cardarelli, L. et al., mBio 2015)

Overstatement of results "these results indicate that diversifying selection within the N-termini of TgrB1 and TgrC1 proteins [...] is the major driver of cooperative aggregation [...]" (p. 5).

In addition, if TgrB1 and TgrC1 binding determines segregation, the co-immunoprecipitation experiments could be used to test binding of proteins with swapped N-termini to substantiate the authors' conclusion here.

The following statement should be clarified "Importantly, this necessarily implies that the degree of binding cannot be a direct function of sequence divergence, since such a linear relationship would not allow for the non-transitive swapping in relative binding. Together these results show that Tgr protein polymorphism can explain diverse partner specific recognition driven by protein binding strength, while providing the specificity required for the locus to act as a polychromatic greenbeard system". This experiment is performed in a cell-free environment and so it is only the protein sequence that can determine binding strength of the proteins. I wonder how else the authors would explain the results if it is not a function of sequence divergence. More explanation is necessary.

'... segregation is mostly eliminated [...] on a non-natural agar substrate' (p. 6). Why is that the case? What happens on a non-natural agar substrate that makes the greenbeard system obsolete? This seems to be an essential finding, as it would show that the greenbeard system is only functional under special conditions. Provide additional information here and discuss this point in more detail.

'...the TgrB1/TgrC1 system [...] represents the first demonstration ...' (p. 7). The authors should be careful with such priority claims.

Figure 2A:

Standard deviation is not shown for the degree of segregation so it is not clear how the statistics were calculated. This is especially striking since the strains that the authors use to show the non-transitive swap have similar segregation values (47% versus 53% and 50% versus 41%).

Figure 2D:

Standard deviation for the blots and biological (not technical replicates) are required. Again the strains that the authors use to show the non-transitive swap have pretty similar strength of binding (29% versus 38% and 49% versus 41%).

Reviewer #3 (Remarks to the Author):

The authors explore the question of whether or not the TgrB1/C1 allorecognition system in *D. discoideum* represents a polychromatic greenbeard that may function to stabilise the cell cooperation that occurs during the aggregative development of social amoebae. This study represents a major advance in this area since they demonstrate the criticality of the tgrB/C locus through whole genome identification of sequence signatures of selection, demonstrate that segregation between wild strains correlates with the relatedness of their tgrB/C loci, and demonstrate differential binding between the ectodomains of TgrB1 and TgrC1 derived from wild isolates. It is their holistic, "bottom-up" approach that greatly strengthens the argument that TgrB1/C1 allorecognition is the main determinant of the evolutionary stabilization of cell cooperation in this system.

Comments that can be corrected by judicious editing of the manuscript follow.

Page 3, bottom paragraph; page 4, bottom of first full paragraph.

The authors make the strong argument for significance of their findings based on the co-occurrence of the wild strains collected by the Francis lab (ref. 25). Since so much of their argument relies on co-occurrence of the strains, more detail on where the strains were collected seems in order, rather than forcing the reader to look it up (I tried online, but did not want to pay for the article). Were the strains from the same square metre?

The brief description of previous work is a little more nuanced than suggested by the text. In Ref 22, half of the isolates were from the same transect at Mountain Lake, Virginia, and so were co-occurring (note that the geographic coordinates provided in Table S1). In Ref. 21, the *tgrB1/C1* loci from three of those co-occurring Virginia strains were moved into an isogenic background and pairwise combinations were found to segregate from one another. So three diverged *tgrB1/C1* loci from three co-occurring wild isolates were found to be necessary and sufficient for the segregation behavior. In the strictest sense the work reported in Ref. 21 demonstrated that TgrB1 and TgrC1 represent a polychromatic greenbeard. The authors' impressive comprehensive analysis of this allorecognition system are not diminished by the previous work, so the authors may wish to reconsider how the previous work is described. The authors draw an artificial distinction between their work and what was demonstrated with 'genetically manipulated' strains in Ref. 21. Instead they should incorporate the findings in Ref. 21 as a genetic demonstration in favour of their argument - perhaps in the middle of the final paragraph?

The experiment described in Figure 2.

It is quite impressive that differential binding of the TgrB1 and TgrC1 ectodomains apparently recapitulates the segregation behaviour of the strains from which they were derived. Since this experiment is carried out with bacterially-produced proteins, without natural glycosylation, and in conditions that may be considered an 'infinite' dilution relative to their occurrence on the amoebal cell surface, the experiment requires a bit more explanation. How many different concentrations of the proteins were tried? How do the authors know they are at non-saturating conditions for binding (would higher concentrations of protein change the observed binding)? How many replications were carried out? Were the Westerns developed on film or by a digital detector? Was saturation of the Western excluded by dilution of the high-concentration bands? I would not require this for publication, but competition experiments with un-tagged versions of the proteins would seem to be the best way to demonstrate the hierarchy. Homotypic partners should outcompete all heterotypic combinations of B1 and C1.

Minor comment:

Last line of 'Results and Discussion'. The sentence should be reworked since a "concept" cannot "operate". Perhaps simply: "...first demonstration of a polychromatic greenbeard operating in a natural system"?

Reviewer #1

I agree with how the authors have interpreted their findings, i.e. that this is a polychromatic greenbeard. However, I think there needs to be a bit more care with regard to the claims for novelty and setting the study in its proper context within the existing literature. In particular, bacteriocins have been conceptualised in terms of the greenbeard effect - e.g. by Gardner et al (2004, Proc B 271: 1529), Gardner & West (2010, reference 12 of this ms), West & Gardner (2010, reference 13 of this ms) and Inglis et al (2011, Am Nat 178: 276) - and models of the evolutionary maintenance of bacteriocin diversity have a 'chromodynamics' flavour - e.g. see Biernaskie et al (2013, J Evol Biol 26: 2081), and references therein - even if this particular term hasn't always been employed. The bacteriocin scenario is different from the present cell-adhesion scenario in that it is harming rather than helping, so perhaps a distinction could be made on that basis, but this would need to be spelled out.

We have now added references to these examples (as well as others highlighted by Reviewer 2). We have also added explicit text to highlight bacteriocin diversity and how this can be seen as a system akin to a harming greenbeard.

Because the requirements for a locus to act as a greenbeard are very restrictive ¹ there have been relatively few reports of greenbeard genes. Examples of putative

greenbeard loci that fulfil some criteria have, however, been described in diverse organisms. These regulate a wide range of recognition phenomena, thus hinting at their potential evolutionary utility and conservation. Putative helping greenbeards include D. discoideum csA¹⁶, N. crassa doc-1, doc-2, doc-3¹⁷; U. stansburiana OBY¹⁸, budding yeast FLO1¹⁹, the B. schlosseri FuHC locus²⁰, P. mirabilis idsD and idsE²¹⁻²³ and the vertebrate Major Histocompatibility Complex²⁴⁻²⁶. Putative harming greenbeards have also been described, including the fire ant Gp9 locus²⁷ and loci controlling bacteriocin production and immunity in bacteria²⁸. Interestingly, bacteriocins are highly polymorphic, with strains also often producing several different bacteriocins. Bacteriocins are thus able to convey highly specific and complex protection against other strains and could therefore be considered polychromatic, albeit regulating harming behaviour^{8,14,28-30}. However, for most putative helping greenbeards, there is little evidence that any of the reported genes can explain complex natural variation in partner-specific patterns of engagement in cooperative behaviours as envisioned by Hamilton¹ and as often observed in nature. For example, even though sequence variation has been described for budding yeast FLO1, which modulates cell-cell adhesion and flocculation (a nominally cooperative trait)¹⁹, this variation simply determines whether cells are competent to cooperate (i.e. 'green' enough). Similarly, although sequence variation and idsD/idsE protein binding in P. mirabilis²¹⁻²³ can explain partner specific interactions during swarming, the extent of allelic variation and thus capacity for complex partner specific interactions is unknown.

2. Page 2, "... an abstract scenario in which... a metaphorical green beard". In Dawkins's (1976) scenario, it's a literal green beard, not a metaphorical one.

The term 'metaphorical' has been deleted.

3. Page 2, "... eliminating its role as an extant modulator of variation in cooperative behaviour" - Gardner & West (2010) suggested that this may render greenbeards relatively resistant to discovery.

Citation has been added, as well as text highlighting that this feature would make such genes difficult to discover.

Because of its obvious fitness benefits, if a discrete greenbeard gene were to emerge it would be expected to rapidly sweep to fixation. Consequently, all individuals would display the greenbeard, leaving the signal devoid of information content⁶⁻⁹, thus eliminating its role as an extant modulator of variation in cooperative behaviour and potentially rendering greenbeard genes relatively resistant to discovery⁸

4. Page 2, the term "falsebeard" was coined by Gardner & West (2010).

Gardner & West (2010) and associated paper citations have been added.

Finally, in a real biological system, falsebeard cheating genotypes would be expected to emerge that display the greenbeard phenotype but do not cooperate^{8,10-12}.

5. Page 4, middle paragraph, it may help the reader to point out explicitly that the greenbeard effect is conceptually distinct from genetic kin discrimination, as these two have often been confused and conflated in the literature. Dawkins (1982, The extended phenotype) and Gardner & West (2010, especially figure 1) spell out the difference.

The idea of distinguishing a greenbeard gene from kin recognition, based on the suggested references is now explicitly spelled out. We have also now extended these ideas into our analysis of the patterns of segregation, their relationship to genome wide variation and thus our conclusion that the *tgrB1/tgrC1* locus fulfils the criteria to be considered a polychromatic greenbeard.

*If segregation is driven by a kin recognition process, then the degree of segregation would be expected to reflect overall genetic distance between strains. This is because common ancestry causes, on average, a similar degree of allele sharing across the whole genome. Therefore, a mechanism based on kin recognition would be expected to show a uniform relationship between allele sharing across the genome and segregation (i.e. the average distance for the whole genome should be predictive of behaviour) ⁴². To test this, we carried out whole genome sequencing of these strains. Hierarchical clustering of 30,444 SNPs revealed a strong phylogenetic signal between the strains (Figure 1B). However, overall genetic distance between strains does not predict the degree of segregation (Figure 1C) ($r = 0.09$, $p = 0.2$). We therefore tested whether segregation behaviour in *D. discoideum* could instead be driven by a polychromatic greenbeard mechanism. For a gene (or genes) to underlie a greenbeard mechanism, we would expect two criteria to be fulfilled. Firstly, a greenbeard locus should exhibit a significant correlation between the allelic similarity of strains and their degree of segregation ⁸. Secondly a candidate polychromatic greenbeard locus must also harbour a level of functional variation that is sufficient to provide the necessary specificity to pairwise interactions underlying segregation. To test whether any genes fulfilled these criteria, we firstly implemented a genome-wide association analysis to test whether the pairwise protein distances between strains (using a set of 6,532 genes with at least one non-synonymous polymorphism) (Supplementary Figure 3, Supplementary Figure 4A and Supplementary Table 1) correlate with the pairwise segregation values. Considering the tests for the individual genes as being approximately independent, we set a threshold based on a Bonferroni correction (using the Šidák equation) with a familywise error rate of 5%, which corresponds to a p-value threshold of 7.85×10^{-6} . This analysis identified 86 greenbeard candidate genes that surpass this significance threshold, with the four largest correlations occurring for genes that map to the same local chromosomal region (Supplementary Figure 4B and Supplementary Table 1). Secondly, we found that seven of the ten most polymorphic of these candidate genes (in terms of number of alternative functional alleles, which in this case corresponds to nine or more alleles) all co-locate to that same genomic region, strongly implicating that region as containing a candidate greenbeard locus. Most strikingly, two genes within that region, *tgrB1* and *tgrC1*, stand out because they top the list of candidates based on each of the two criteria. They have the two highest levels of allelic diversity among the candidates (with 18 alleles each out of 20 strains) and the highest levels of total*

sequence variation (synonymous and non-synonymous) in the genome (Supplementary Figure 4C and Supplementary Table 1), while also showing the two largest correlations with segregation (see Supplementary Table 1 and Supplementary Figure 5 for examples of segregation correlation for other highly polymorphic genes). Critically, although our analysis identifies two candidate genes that show very tight physical linkage (i.e., they are only ca. 500bp apart) and functional biochemical coupling^{37,38,41}, and therefore the pair can constitute a single locus (the "tgr locus"), as required by the greenbeard mechanism. Considering the entire functional tgr locus sequence, we find that there is sufficient allelic diversity for every strain to carry a unique allele, thus providing sufficient putative variation to explain the patterns of segregation observed. Further strong support for the tgr locus as a polychromatic greenbeard candidate comes from gene knockout and gene swapping experiments in isogenic laboratory strains, which have shown that a matching pair of tgrB1 and tgrC1 alleles is necessary and sufficient for attractive self-recognition and cell-cell adhesion⁴¹.

Reviewer #2

The significance of the study paper would be substantially increased by adding additional biochemical data on the hypothesis that the strength of protein binding is the underlying cause of the "polychromatic" greenbeard effect. The data as they are currently presented is preliminary. An example of assessing physical interaction of adhesion proteins has been biochemically characterized by the Gibbs laboratory (see below). Data supporting the hypothesis that binding activity of the Tgr proteins is the biochemical basis of the "polychromatic" aspect of these proteins is not fully substantiated by the data presented in this paper.

As suggested we have now added further biological and technical replication to support the biochemical data. The data are now drawn from three protein preparations (which are reproducible when separated by 2 years), for each strain pairing the patterns are drawn from at least 6 biological replicates and we have carried out the pull down using a >100x range of different protein concentrations. These data are now described in modified figure 2 and new supplementary figure 11. Consequently we feel that the relationships and conclusions drawn from the biochemical data are now better substantiated.

*We next tested whether Tgr protein interaction strength could predict segregation behaviour. Indeed, previous studies have shown that Tgr proteins mediate cell-cell interactions^{39,40}, whilst TgrB1 and TgrC1 interactions are required for clustering and adhesion complex formation⁴¹. Moreover, specific regions have been defined that are required for in vitro protein interactions when isolated from *D. discoideum* extracts³⁸. However, because the vast majority of highly variable sites do not occur in regions previously described as being required for protein-protein interaction (Supplementary Figures 8-10), we expressed almost full-length TgrB1 and TgrC1 proteins from these strains in bacteria (amino acids 65-861 and 57-867 respectively or 88.47% and 98.5% of each coding sequence, see methods for details). Each protein thus contained the domains previously described*

to mediate protein interactions, as well as the highly polymorphic N-terminal region of unknown function. In fact the sequence within the known binding domain and C-terminal regions is actually identical for TgrB1 in three of the chosen strains (Supplementary figures 9 and 10 for full alignments and primer positions). The strength of protein interactions was tested in pairwise co-immunoprecipitation experiments (Figure 2B). In all cases, the strength of binding (Figure 2D) strongly correlates with the degree of segregation (Figure 2C) and even predicts the non-transitive swap (Figure 2D). It is important to note that binding patterns were reproducible over a 125-fold range of protein concentrations (Figure 2C and Supplementary Figure 11). Therefore, even though Tgr protein concentrations on the cell surface are unknown, binding patterns are extremely robust. Furthermore, the observed binding patterns necessarily imply that the degree of binding cannot simply be a direct function of the number of SNPs, since such a linear relationship would not allow for the non-transitive swapping in relative binding. Finally, since the TgrB1 sequence within the known binding domain and C-terminal regions is identical in three of the tested strains (Supplementary Figure 9), taken together, these results thus strongly suggest that precise combinations of Tgr protein polymorphism within the N-terminus can explain diverse partner-specific recognition driven by protein binding strength, while providing the specificity required for the locus to act as a polychromatic greenbeard system.

The introduction to the paper describes the greenbeard effect and introduces the theory underlying this phenomenon. However, the authors should cite some additional literature outside of the *D. discoideum* field. They claim that there have been relatively few reports of potential greenbeard genes, although reports of greenbeard genes have existed for at least 15 years (see Keller and Ross (1998) *Nature*; as an example). In a recent study of the filamentous fungus *Neurospora crassa*, a 'polychromatic' greenbeard example has been characterized in detail (Heller et al. (2016) *PLoS Biology*). Other examples for polymorphic genes that mediate non-self recognition and could be considered greenbeard genes are described in *B. schlosseri* (see work of the De Tomaso group) and *Proteus mirabilis* (see work of the Gibbs group). These studies should be included in the introduction in addition to the budding yeast *FLO1* because they show that the greenbeard effect is found in a wide group of organisms and therefore is of interest for a broader audience.

This introduction has been expanded to include this helpful suggestion.

*Because the requirements for a locus to act as a greenbeard are very restrictive ¹ there have been relatively few reports of greenbeard genes. Examples of putative greenbeard loci that fulfil some criteria have, however, been described in diverse organisms. These regulate a wide range of recognition phenomena, thus hinting at their potential evolutionary utility and conservation. Putative helping greenbeards include *D. discoideum* *csA* ¹⁶, *N. crassa* *doc-1*, *doc-2*, *doc-3* ¹⁷; *U. stansburiana* *OBY* ¹⁸, budding yeast *FLO1* ¹⁹, the *B. schlosseri* *FuHC* locus ²⁰, *P.**

mirabilis *idsD* and *idsE* ²¹⁻²³ and the vertebrate Major Histocompatibility Complex ²⁴⁻²⁶. Putative harming greenbeards have also been described, including the fire ant *Gp9* locus ²⁷ and loci controlling bacteriocin production and immunity in bacteria ²⁸. Interestingly, bacteriocins are highly polymorphic, with strains also often producing several different bacteriocins. Bacteriocins are thus able to convey highly specific and complex protection against other strains and could therefore be considered polychromatic, albeit regulating harming behaviour ^{8,14,28-30}. However, for most putative helping greenbeards, there is little evidence that any of the reported genes can explain complex natural variation in partner-specific patterns of engagement in cooperative behaviours as envisioned by Hamilton ¹ and as often observed in nature. For example, even though sequence variation has been described for budding yeast *FLO1*, which modulates cell-cell adhesion and flocculation (a nominally cooperative trait) ¹⁹, this variation simply determines whether cells are competent to cooperate (i.e. 'green' enough). Similarly, although sequence variation and *idsD/idsE* protein binding in *P. mirabilis* ²¹⁻²³ can explain partner specific interactions during swarming, the extent of allelic variation and thus capacity for complex partner specific interactions is unknown. Despite this, both examples highlight the fact that cell adhesion proteins represent a standout candidate to encode a polychromatic greenbeard ³¹. This is because they are localised at the cell surface, giving cells the ability to differentially adhere to other cells expressing the same molecule and thus can directly modulate cell behaviours ³¹. Crucially, sequence variation could potentially generate a spectrum of beard colours, and thus provide the necessary specificity for identifying compatible partners. Here we test this conjecture and demonstrate that a variable cell adhesion system represents a polychromatic greenbeard that underlies variation and co-existence in cooperative behaviour in a natural population of *D. discoideum*.

Additional information on the model for Tgr function and past biochemical characterization should be explicitly described in the paper.

Text and associated references have been added to explicitly describe studies where biochemical characterization of Tgr proteins has been carried out. We have also highlighted the relationship of the domains characterized in these studies to the polymorphisms described here in new supplementary figures and accompanying legends.

We next tested whether Tgr protein interaction strength could predict segregation behaviour. Indeed, previous studies have shown that Tgr proteins mediate cell-cell interactions ^{39,40}, whilst TgrB1 and TgrC1 interactions are required for clustering and adhesion complex formation ⁴¹. Moreover, specific regions have been defined that are required for *in vitro* protein interactions when isolated from *D. discoideum* extracts ³⁸. However, because the vast majority of highly variable sites do not occur in regions previously described as being required for protein-protein interaction (Supplementary Figures 8-10), we expressed almost full-length TgrB1 and TgrC1 proteins from these strains in bacteria (amino acids 65-861 and 57-867

respectively or 88.47% and 98.5% of each coding sequence, see methods for details).

The authors want to test if strains are able to choose their partners in chimeras (p. 3), but then only test segregation in pairwise mixtures. It would be interesting to see if segregation can be confirmed if more than two strains are mixed. For example, in a mixture of three strains (two with strong affinity, one with weak affinity), there should be a clear difference in segregation. The two strains with strong affinity should not segregate and appear in the same FBs while the strain with weak affinity to the other strains should segregate.

The 3-way experiment has been performed as suggested by the Reviewer. Results are as predicted by 2-way mixes (i.e. strain preferences do not change in 3-way mixes). Two examples are described in the text, methods added and data are shown in new supplementary figure 2 and accompanying legend.

Most importantly, strains exhibit considerable diversity in partner-specific segregation (Figure 1A), with 3-way mixes also following expectations based on segregation behaviour in pairwise interactions (Supplementary Figure 2).

The authors sequence 20 wild isolates of *D. discoideum* and map the sequence back to the reference strain. Is it possible that additional loci show significant polymorphism between the strains and the reference sequence due to an inability to map back to the genome sequence of the reference strain? Was de novo assembly of any of the 20 genomes performed to test this hypothesis?

De novo assemblies of coding sequences for each strain were conducted to verify SNPs especially in the tgr genes. Assembled sequences were compared to sequences derived from mapping to ascertain that genes with very high numbers of SNPs weren't overlooked due to reads not mapping or wrong SNPs were called due to reads mapping to the wrong gene. This is now described in the methods ('Assembly of nucleotide sequences').

Assembly of nucleotide sequences

*To obtain sequences of genes showing a high number of filtered SNPs, trimmed reads were assembled using velvet Version 1.2.10⁶³⁻⁶⁵ and k-mer values between 21 and 85. Contigs from all assemblies were then searched against a database of 13,409 known *D. discoideum* transcripts downloaded from dictyBase⁶⁶ at 02.05.2013 using BLAST version 2.2.27 (parameters: blastn, E-value $\leq 10^{-5}$; (Altschul, 1997 #50)). All sequences with either tgrB1 or tgrC1 as best hit were further assembled using CAP3⁶⁷ and then manually corrected using CLC Genomics workbench 6.*

In addition, SNPs within the tgrB1 and tgrC1 were further validated by Sanger sequencing. 'Sanger sequencing of tgr genes' has been added to methods.

Sanger sequencing of tgr genes

SNPs in highly polymorphic regions of tgrB1 and tgrC1 were further validated by Sanger sequencing (tgrC1 forward primer: GAACCCAGAACTGAAATGGCAC; tgrC1 reverse primer: GTAATAGGCAAGAGCACC; tgrB1 forward: CAATATTAGTAGTAGTGGGATTC; tgrB1 reverse: CCGAAACCAGGTCCTAGAAC). Primer locations are highlighted in Supplementary Figures 9 and 10. All SNPs called from the mapping data in these regions were validated in the Sanger sequences.

Many microbes contain a number of loci that are highly polymorphic and that show diversifying selection. Is the Tgr locus the only identified locus?

This idea is now addressed extensively in the manuscript and in accompanying supplementary material. For example, we have included a supplementary table describing all polymorphic genes and their relationship to segregation behavior and whole genome evolution. This information is also summarized in new supplementary figures 3-5.

*If segregation is driven by a kin recognition process, then the degree of segregation would be expected to reflect overall genetic distance between strains. This is because common ancestry causes, on average, a similar degree of allele sharing across the whole genome. Therefore, a mechanism based on kin recognition would be expected to show a uniform relationship between allele sharing across the genome and segregation (i.e. the average distance for the whole genome should be predictive of behaviour)⁴². To test this, we carried out whole genome sequencing of these strains. Hierarchical clustering of 30,444 SNPs revealed a strong phylogenetic signal between the strains (Figure 1B). However, overall genetic distance between strains does not predict the degree of segregation (Figure 1C) ($r = 0.09$, $p = 0.2$). We therefore tested whether segregation behaviour in *D. discoideum* could instead be driven by a polychromatic greenbeard mechanism. For a gene (or genes) to underlie a greenbeard mechanism, we would expect two criteria to be fulfilled. Firstly, a greenbeard locus should exhibit a significant correlation between the allelic similarity of strains and their degree of segregation⁸. Secondly a candidate polychromatic greenbeard locus must also harbour a level of functional variation that is sufficient to provide the necessary specificity to pairwise interactions underlying segregation. To test whether any genes fulfilled these criteria, we firstly implemented a genome-wide association analysis to test whether the pairwise protein distances between strains (using a set of 6,532 genes with at least one non-synonymous polymorphism) (Supplementary Figure 3, Supplementary Figure 4A and Supplementary Table 1) correlate with the pairwise segregation values. Considering the tests for the individual genes as being approximately independent, we set a threshold based on a Bonferroni correction (using the Šidák equation) with a familywise error rate of 5%, which corresponds to a p-value threshold of 7.85×10^{-6} . This analysis identified 86 greenbeard candidate genes that surpass this significance threshold, with the four largest correlations occurring for genes that map to the same local chromosomal region (Supplementary Figure 4B and Supplementary Table 1). Secondly, we found that seven of the ten most polymorphic of these candidate genes (in terms of number of alternative functional alleles, which in this case corresponds to nine or more alleles) all co-locate to that same genomic region, strongly implicating that region*

as containing a candidate greenbeard locus. Most strikingly, two genes within that region, tgrB1 and tgrC1, stand out because they top the list of candidates based on each of the two criteria. They have the two highest levels of allelic diversity among the candidates (with 18 alleles each out of 20 strains) and the highest levels of total sequence variation (synonymous and non-synonymous) in the genome (Supplementary Figure 4C and Supplementary Table 1), while also showing the two largest correlations with segregation (see Supplementary Table 1 and Supplementary Figure 5 for examples of segregation correlation for other highly polymorphic genes). Critically, although our analysis identifies two candidate genes that show very tight physical linkage (i.e., they are only ca. 500bp apart) and functional biochemical coupling^{37,38,41}, and therefore the pair can constitute a single locus (the “tgr locus”), as required by the greenbeard mechanism. Considering the entire functional tgr locus sequence, we find that there is sufficient allelic diversity for every strain to carry a unique allele, thus providing sufficient putative variation to explain the patterns of segregation observed. Further strong support for the tgr locus as a polychromatic greenbeard candidate comes from gene knockout and gene swapping experiments in isogenic laboratory strains, which have shown that a matching pair of tgrB1 and tgrC1 alleles is necessary and sufficient for attractive self-recognition and cell-cell adhesion⁴¹.

The authors claim that 'for the vast majority of these genes, the genetic distance between strains mirrors the pattern of overall distance', (p. 4). Does vast majority of genes mean that there are other loci beside the tgr locus that show a larger correlation with segregation behavior than the genome-wide pattern? If so, the authors should mention these loci and explain why they focused on the Tgr locus.

We have significantly altered the presentation in this section to emphasize that a potential polychromatic greenbeard gene needs to fulfill two essential criteria – it must significantly correlate with behavior while also harboring sufficient variation to allow for the specificity of pairwise interactions observed (see the paragraph included above in response to the previous comment). Both of these criteria independently and strongly implicate the tgr genes as the most likely candidates.

Briefly, in our presentation we now firstly identify the list of genes that show a significant correlation with segregation. For this criterion, it is important to note that it is not sufficient for a gene to show a pattern larger than that seen for the genome-wide pattern, since we expect this to be true for a large fraction of the genome (since the null expectation is for the correlation to randomly vary across loci, with the genome wide pattern simply being the average of the individual loci), it must show a correlation that is larger than that expected by random chance. Of the genes that show a correlation that exceeds the significance threshold, the most correlated genes co-locate to the tgr locus, and the tgr genes top that list with two highest correlations. Secondly, that same region also shows the highest allelic diversity among the candidates (again with the tgr genes topping the list). Thus, we focus on the tgr locus because evidence from both essential criteria point to the same genomic region, with the tgr genes standing out atop the candidates.

In addition, a plot comparing the relationship between tree distance and segregation values for several of the identified polymorphic loci could be included as a control.

This suggestion is now incorporated as suggested for several of the most polymorphic genes in a new Supplementary Figure 5. We have also addressed the issue of 'as a control' by using a significance threshold to filter the candidates (see above).

Most strikingly, we found that two genes within that region, tgrB1 and tgrC1, stand out. Not only do they have the two highest levels of allelic diversity among the candidates (with 18 alleles each out of 20 strains) and the highest levels of total sequence variation (synonymous and non-synonymous) in the genome (Supplementary Figure 4C and Supplementary Table 1), they also show the two largest correlations with segregation (see Supplementary Table 1 and Supplementary Figure 5 for examples of segregation correlation for other highly polymorphic genes).

'...supporting the idea that the genes show concerted evolution as a single functional locus' (p. 4). A comparative phylogenetic analysis for each gene would help to visualize that the tree structure is indeed the same.

This suggestion has now been incorporated. This analysis highlights the fact that the tree structure each gene is similar, but not identical. This is highlighted in the text and new Supplementary Figure 6.

...analyses of these final gene sequences revealed that, while the phylogenetic trees computed from the variants present at these genes are not identical (Supplementary Figure 6), distances between strains of both tgr genes are very highly positively correlated ($r = 0.89$; $p = 1.84 \times 10^{-128}$; Supplementary Figure 7A), supporting the idea that the genes show concerted evolution as a single functional locus.

Co-immunoprecipitation experiments: In the materials and methods, it is indicated that "a fragment of TgrB1 from each isolate was amplified by PCR...". What fragment? How was this region chosen? This information is not sufficient to evaluate whether the methods and results are accurate for the co-immunoprecipitation experiments. Additionally, this section of the paper should be expanded with additional experiments to support the contention that variation in binding of the Tgr protein is the biochemical basis of the "polychromatic" aspect of these proteins. The authors should look at the paper from the Gibbs lab on the characterization of adhesion proteins that mediate self/nonself interaction in *Proteus* for examples (Cardarelli, L. et al., mBio 2015)

We feel that this confusion comes from our poor explanation and use of the word 'fragment'. The constructs resulted in almost full length proteins. We have now made this point clearly in the text and new Supplementary figures 9 and 10,

which describe the precise details as suggested. Consequently, the *tgrB1* proteins expressed are even equivalent to the domain swaps of the type used in Cardarelli et al. In addition, as suggested we have now added further biological and technical replication to the support biochemical data. The data is now drawn from three protein preparations (which are reproducible when separated by 2 years), for each strain pairing the patterns are drawn from at least six biological replicates and we have carried out the pull down using a >100 fold range of different protein concentrations. These data are now described in modified figure 2 and a new supplementary figure. Consequently we feel that the relationships and conclusions drawn from the biochemical data are now greatly substantiated.

We next tested whether Tgr protein interaction strength could predict segregation behaviour. Indeed, previous studies have shown that Tgr proteins mediate cell-cell interactions^{39,40}, whilst TgrB1 and TgrC1 interactions are required for clustering and adhesion complex formation⁴¹. Moreover, specific regions have been defined that are required for in vitro protein interactions when isolated from D. discoideum extracts³⁸. However, because the vast majority of highly variable sites do not occur in regions previously described as being required for protein-protein interaction (Supplementary Figures 8-10), we expressed almost full-length TgrB1 and TgrC1 proteins from these strains in bacteria (amino acids 65-861 and 57-867 respectively or 88.47% and 98.5% of each coding sequence, see methods for details). Each protein thus contained the domains previously described to mediate protein interactions, as well as the highly polymorphic N-terminal region of unknown function. In fact the sequence within the known binding domain and C-terminal regions is actually identical for TgrB1 in three of the chosen strains (Supplementary figures 9 and 10 for full alignments and primer positions). The strength of protein interactions was tested in pairwise co-immunoprecipitation experiments (Figure 2B). In all cases, the strength of binding (Figure 2D) strongly correlates with the degree of segregation (Figure 2C) and even predicts the non-transitive swap (Figure 2D). It is important to note that binding patterns were reproducible over a 125-fold range of protein concentrations (Figure 2C and Supplementary Figure 11). Therefore, even though Tgr protein concentrations on the cell surface are unknown, binding patterns are extremely robust. Furthermore, the observed binding patterns necessarily imply that the degree of binding cannot simply be a direct function of the number of SNPs, since such a linear relationship would not allow for the non-transitive swapping in relative binding. Finally, since the TgrB1 sequence within the known binding domain and C-terminal regions is identical in three of the tested strains (Supplementary Figure 9), taken together, these results thus strongly suggest that precise combinations of Tgr protein polymorphism within the N-terminus can explain diverse partner-specific recognition driven by protein binding strength, while providing the specificity required for the locus to act as a polychromatic greenbeard system.

Overstatement of results "these results indicate that diversifying selection within the N-termini of TgrB1 and TgrC1 proteins [...] is the major driver of cooperative aggregation [...]' (p. 5). In addition, if TgrB1 and TgrC1 binding determines segregation, the co-immunoprecipitation experiments could be used to test binding of proteins with swapped N-termini to substantiate the authors' conclusion here.

We agree with this suggestion. In fact, Nature has performed this domain swap for us. For three strains tested, the tgrB1 protein sequence is actually identical outside the N-terminus (including invariant sequence in the known protein interaction domain). Despite this, each protein exhibits quite different protein interaction strengths when compared against each tgrC1 protein. This important point is now made in the text and highlighted in new Supplementary Figures 9 and 10.

Each protein thus contained the domains previously described to mediate protein interactions, as well as the highly polymorphic N-terminal region of unknown function. In fact the sequence within the known binding domain and C-terminal regions is actually identical for TgrB1 in three of the chosen strains (Supplementary figures 9 and 10 for full alignments and primer positions).

and

Finally, since the TgrB1 sequence within the known binding domain and C-terminal regions is identical in three of the tested strains (Supplementary Figure 9), taken together, these results thus strongly suggest that precise combinations of Tgr protein polymorphism within the N-terminus can explain diverse partner-specific recognition driven by protein binding strength, while providing the specificity required for the locus to act as a polychromatic greenbeard system.

The following statement should be clarified "Importantly, this necessarily implies that the degree of binding cannot be a direct function of sequence divergence, since such a linear relationship would not allow for the non-transitive swapping in relative binding. Together these results show that Tgr protein polymorphism can explain diverse partner specific recognition driven by protein binding strength, while providing the specificity required for the locus to act as a polychromatic greenbeard system". This experiment is performed in a cell-free environment and so it is only the protein sequence that can determine binding strength of the proteins. I wonder how else the authors would explain the results if it is not a function of sequence divergence. More explanation is necessary.

This stems from our imprecise wording that confused our intended meaning. We did not mean to imply that sequence divergence did not underlie changes in activity. Rather we meant that the simple NUMBER of changes could not be the driver, but that the actual amino acid composition is crucial. The text has been modified to avoid this confusion.

Furthermore, the observed binding patterns necessarily imply that the degree of binding cannot simply be a direct function of the number of SNPs, since such a linear relationship would not allow for the non-transitive swapping in relative binding.

'... segregation is mostly eliminated [...] on a non-natural agar substrate' (p. 6). Why is that the case? What happens on a non-natural agar substrate that makes the greenbeard system obsolete? This seems to be an essential finding, as it would show that the greenbeard system is only functional under special conditions. Provide additional information here and discuss this point in more detail.

It has been recognized for many years that *D. discoideum* development is affected by the type of substrate used, as well as many other environmental factors, including light and humidity. In the case of soil vs agar, it is generally acknowledged that soil development is the more natural substrate, and thus reflects the state to which cells are adapted. One reason for this is that many gene knockouts exhibit no phenotype when developed on soil. Under these conditions, genes thus appear redundant. However, these same knockouts often exhibit strong phenotypes when developed on soil. If so, segregation and greenbeard behavior is likely the normal mode of development, rather than a 'special condition'. The reason behind differences on soil and agar is unknown (and could be hugely multifactorial). These differences have, however, been speculated to stem from soil development resulting in a more challenging physiological environment for cell migration. Although further studies to better characterise these differences would indeed be interesting, we do not, however, feel that it falls within the scope of this study. Moreover, it does not affect the observations or conclusions. Nevertheless, as per the reviewer's helpful suggestion, we have now added a discussion of soil vs agar development to the text, as well as references to the studies described above.

For this, we exploited the fact that segregation is mostly eliminated when strains undergo development on a non-natural agar substrate (Supplementary Figure 12). The reason for this is unknown, but development on soil is thought to result in a more physiologically relevant and challenging physical environment for cell behaviours such as cell migration, which is important during aggregation. Indeed, soil development has previously been shown to uncover phenotypic defects caused by null mutations that are silent on agar development^{16,50}. Hence, we were able to measure the consequences of chimeric development for pairs of strains that would naturally avoid chimerism through segregation.

'...the TgrB1/TgrC1 system [...] represents the first demonstration ...' (p. 7). The authors should be careful with such priority claims.

We have removed all priority claims, as suggested

Figure 2A: Standard deviation is not shown for the degree of segregation so it is not clear how the statistics were calculated. This is especially striking since the

strains that the authors use to show the non-transitive swap have similar segregation values (47% versus 53% and 50% versus 41%).

Error bars are now shown for the segregation values in Figure 2C. Bootstrapped t-tests were used to confirm that differences between all mixes were statistically significant. These methods are described in the 'Statistical analyses and tests for selection' section of the methods. We have also now stated in the legend to Figure 2A that although the segregation values of some pairing are indeed quite similar, thorough statistical tests confirm that they are indeed significantly different.

Strains NC96.1 and NC34.2 have the highest segregation values of these mixes (64%). Strain NC71.1 segregates the least from both strain NC34.2 (47%) and NC96.1 (41%), while strain NC52.3 segregates more from both strains (53% and 50%). Red arrows depict the non-transitive swap of strains NC52.3 and NC71.1, where strains NC34.2 and NC96.1 both segregate less from NC71.1. Mean segregation values of all pairings within each hierarchy are significantly different (t-test; $p < 2.2 \times 10^{-16}$).

Figure 2D: Standard deviation for the blots and biological (not technical replicates) are required. Again the strains that the authors use to show the non-transitive swap have pretty similar strength of binding (29% versus 38% and 49% versus 41%).

We have added text to the legend for Figure 2C which now better describes how the error bars were generated from biological replicates. We have also added statistics to the legend for Figure 2D that highlight that differences in binding strengths of each pairing are sometimes small, yet the differences are highly statistically significant.

For each pairwise strain comparison, normalised segregation values were plotted against percent binding. Because protein concentration does not affect binding (Supplementary Figure 11), interactions at four different protein concentrations were considered technical replicates and used to calculate the mean for each biological replicate. A minimum of six biological replicates were performed for each protein pairing. Y-axis error bars depict standard error of the mean percentage of bound protein in each fraction of the biological replicates, quantified by measuring band intensities on western blots using ImageJ⁸¹. X-axis error bars depict standard error of the mean normalised segregation value derived from 1000 bootstrapped samples (see methods). Blue line depicts highly negative and significant ($r = -0.98$, $p = 0.0004$) correlation between segregation values and protein binding for mixes of four chosen strains (gray shading indicating the 99% confidence envelope). Self mixes, which were used as controls, are excluded.

Tgr proteins from NC96.1 and NC34.2 exhibit the lowest binding (14%). Tgr proteins from NC34.2 bind more strongly to NC71.1 (38%) than NC52.3 (28%).

Similarly, Tgr proteins from NC96.1 also bind more strongly to NC71.1 (49%) than NC52.3 (41%). Red arrows depict the non-transitive swap of strains NC52.3 and NC71.1, where proteins of strains NC34.2 and NC96.1 both bind more to Tgr protein from NC71.1. This pattern mirrors the non-transitive behaviour of the segregation values (shown in panel A). Mean binding values between each pairing in the hierarchy are significantly different (one way ANOVA; $p < 2.2 \times 10^{-16}$ and Tukey multiple comparisons of means using adjusted p -value 0.05).

Reviewer #3

The authors make the strong argument for significance of their findings based on the co-occurrence of the wild strains collected by the Francis lab (ref. 25). Since so much of their argument relies on co-occurrence of the strains, more detail on where the strains were collected seems in order, rather than forcing the reader to look it up (I tried online, but did not want to pay for the article).

We now include a more detailed explanation of strain collection and associated references to better describe why the strains chosen are this is widely accepted to be a part of a freely mixing population.

To test this idea, we firstly determined whether naturally co-occurring strains are able to 'choose' their partners in chimera. We measured the degree of segregation within pairwise mixtures of 20 strains isolated from the same North Carolina locale ^{34,35,43-45}. These strains were isolated from 1m² patches of soil and have been shown to exhibit little linkage disequilibrium, suggesting that recombination and mixing is common ^{33,44}.

The brief description of previous work is a little more nuanced than suggested by the text. In Ref 22, half of the isolates were from the same transect at Mountain Lake, Virginia, and so were co-occurring (note that the geographic coordinates provided in Table S1). In Ref. 21, the tgrB1/C1 loci from three of those co-occurring Virginia strains were moved into an isogenic background and pairwise combinations were found to segregate from one another. So three diverged tgrB1/C1 loci from three co-occurring wild isolates were found to be necessary and sufficient for the segregation behavior. In the strictest sense the work reported in Ref. 21 demonstrated that TgrB1 and TgrC1 represent a polychromatic greenbeard. The authors' impressive comprehensive analysis of this allorecognition system are not diminished by the previous work, so the authors may wish to reconsider how the previous work is described. The authors draw an artificial distinction between their work and what was demonstrated with 'genetically manipulated' strains in Ref. 21. Instead they should incorporate the findings in Ref. 21 as a genetic demonstration in favour of their argument - perhaps in the middle of the final paragraph?

We thank the reviewer for this suggestion. We have now included this historical information and clarified how our studies are distinguished from these observations, as suggested.

*Indeed, segregation has been reported between strains, with the degree of segregation shown to correlate to geographic and genetic distance ³⁷. These strains also exhibit high levels of polymorphism at two loci, tgrB1 and tgrC1, which are thought to encode cell adhesion molecules ³⁸⁻⁴⁰. Furthermore, elegant gene swapping experiments have shown that matching tgrB1 and tgrC1 alleles are required for co-aggregation ^{37,41,42}. These studies thus raised the possibility that a polychromatic greenbeard system based on TgrB1 and TgrC1 mediated cell adhesion could underlie cooperative behaviour in natural populations of *D. discoideum*. To test this idea, we firstly determined whether naturally co-occurring*

strains are able to 'choose' their partners in chimera. We measured the degree of segregation within pairwise mixtures of 20 strains isolated from the same North Carolina locale ^{34,35,43-45}.

The experiment described in Figure 2. It is quite impressive that differential binding of the TgrB1 and TgrC1 ectodomains apparently recapitulates the segregation behaviour of the strains from which they were derived. Since this experiment is carried out with bacterially-produced proteins, without natural glycosylation, and in conditions that may be considered an 'infinite' dilution relative to their occurrence on the amoebal cell surface, the experiment requires a bit more explanation. How many different concentrations of the proteins were tried? How do the authors know they are at non-saturating conditions for binding (would higher concentrations of protein change the observed binding)? How many replications were carried out? Were the Westerns developed on film or by a digital detector? Was saturation of the Western excluded by dilution of the high-concentration bands? I would not require this for publication, but competition experiments with un-tagged versions of the proteins would seem to be the best way to demonstrate the hierarchy. Homotypic partners should outcompete all heterotypic combinations of B1 and C1.

We have incorporated all these useful suggestions. As suggested, we have now added further biological and technical replication to the support biochemical data. The data is now drawn from three protein preparations (which are reproducible when separated by 2 years), for each strain pairing the patterns are drawn from at least six biological replicates and we have carried out the pull down using a >100 fold range of different protein concentrations. Because we find that protein concentration does not affect binding, we have considered these as technical replicates. Finally, as the reviewer notes, quantification was carried out following film development. We have therefore carried out a dilution series to ensure that all band quantified are in the linear range. This data are now described in modified figure 2 and a new supplementary figure. Consequently we feel that the relationships and conclusions drawn from the biochemical data are now better substantiated.

We next tested whether Tgr protein interaction strength could predict segregation behaviour. Indeed, previous studies have shown that Tgr proteins mediate cell-cell interactions ^{39,40}, *whilst TgrB1 and TgrC1 interactions are required for clustering and adhesion complex formation* ⁴¹. *Moreover, specific regions have been defined that are required for in vitro protein interactions when isolated from D. discoideum extracts* ³⁸. *However, because the vast majority of highly variable sites do not occur in regions previously described as being required for protein-protein interaction (Supplementary Figures 8-10), we expressed almost full-length TgrB1 and TgrC1 proteins from these strains in bacteria (amino acids 65-861 and 57-867 respectively or 88.47% and 98.5% of each coding sequence, see methods for details). Each protein thus contained the domains previously described to mediate protein interactions, as well as the highly polymorphic N-terminal region of unknown function. In fact the sequence within the known binding domain*

and C-terminal regions is actually identical for TgrB1 in three of the chosen strains (Supplementary figures 9 and 10 for full alignments and primer positions). The strength of protein interactions was tested in pairwise co-immunoprecipitation experiments (Figure 2B). In all cases, the strength of binding (Figure 2D) strongly correlates with the degree of segregation (Figure 2C) and even predicts the non-transitive swap (Figure 2D). It is important to note that binding patterns were reproducible over a 125-fold range of protein concentrations (Figure 2C and Supplementary Figure 11). Therefore, even though Tgr protein concentrations on the cell surface are unknown, binding patterns are extremely robust. Furthermore, the observed binding patterns necessarily imply that the degree of binding cannot simply be a direct function of the number of SNPs, since such a linear relationship would not allow for the non-transitive swapping in relative binding. Finally, since the TgrB1 sequence within the known binding domain and C-terminal regions is identical in three of the tested strains (Supplementary Figure 9), taken together, these results thus strongly suggest that precise combinations of Tgr protein polymorphism within the N-terminus can explain diverse partner-specific recognition driven by protein binding strength, while providing the specificity required for the locus to act as a polychromatic greenbeard system.

Immunoprecipitation of bacterially expressed His₆ - TgrC1 and GST - TgrB1 from four different strains. His₆ - TgrC1 was incubated with GST - TgrB1 from each strain. His₆ - TgrC1 complexes were isolated and any bound GST - TgrB1 was detected with an anti GST antibody (I = input protein, NB = not bound protein and B = bound protein). The strength of binding can be quantified by comparing the relative amount of protein (band intensity) in the bound and not bound fractions. The highest binding was observed between proteins from the same strain. The weakest binding was observed between Tgr proteins from NC34.2 and NC96.1. Varying levels of protein binding were observed between the Tgr proteins from other strains. Importantly, the strength of protein binding between Tgr proteins from the different strains shows the same pattern as segregation between the strains.

B. Quantification of protein binding. The strength of protein binding between Tgr proteins from the different strains is unaffected by protein concentration.

C. All densitometry measurements are in the linear range in all protocols used. Top panel: 25µg of GST-TgrB1 protein was loaded neat, at a 1:2 dilution or a 1:4 dilution or 1:10 dilution and detected by Western blotting using anti-GST antibody. Bottom panel: Densitometry analyses of GST-TgrB1 compared to protein concentration. Error bars represent s.e.m. from three independent GST-TgrB1 protein preparations.

Minor comment: Last line of 'Results and Discussion'. The sentence should be reworked since a "concept" cannot "operate". Perhaps simply: "...first demonstration of a polychromatic greenbeard operating in a natural system"?

We have removed this sentence.

REVIEWERS' COMMENTS:

Reviewer #2 (Remarks to the Author):

In this re-submission the authors have addressed the concerns raised in the initial submission. The manuscript has improved and the results are presented more clearly. The authors extend results on allorecognition obtained previously from elegant experiments in the lab strain to wild isolates and document the finding that the Tgr locus is a polychromatic greenbeard locus. However, since the Tgr locus has been shown previously to be polymorphic and necessary and sufficient for self/nonself recognition during fruiting body formation of *D. discoideum*, the novelty and enthusiasm for this work is somewhat diminished. The first part of the paper (identification of that same locus as being the best candidate for a polychromatic greenbeard locus) could be shortened.

Introduction

The greenbeard effect is described more clearly now in the introduction and the literature cited covers a broader field, which highlights the importance of this phenomenon for a broad readership.

Results and Discussion

The results and discussion section has improved considerably. Some modifications are still necessary:

'Critically, although our analysis identifies two candidate genes that show very tight physical linkage (i.e., they are only ca. 500 bp apart) and functional biochemical coupling and therefore the pair can constitute a single locus (the "tgr locus"), as required by the greenbeard mechanism.' – correct sentence

'Firstly, analyses of these final gene sequences revealed that, while the phylogenetic trees computed from the variants present at these genes are not identical (Supplementary Fig. 6), distances between strains of both tgr genes are very highly positively correlated ($r = 0.89$; $p = 1.84 \times 10^{-128}$; Supplementary Fig. 7A), supporting the idea that the genes show concerted evolution as a single functional locus.' –

The fact that the phylogenetic trees of both tgr genes do not show identical structures suggests that recombination occurs between both genes and that they do not serve as a single functional locus. Genetic recombination occurring within this locus could be tested bioinformatically (for example using RDP, Martin et al (2015) Virus Evolution).

'(tgrB1: $r = 0.24$; $p = 0.001,19$ tgrC1: $r = 0.21$; $p = 0.004$)' – is there a graph/plot showing these results?

Figures

In general the figure captions need better explanations. This is especially true for the supplementary figures.

Supplementary Figure 2:

The numbers in parentheses need better explanation.

Supplementary Figure 8:

The description says allele A and allele B. In supplementary figure 4C it is stated that TgrB1 and TgrC1 have 18 alleles each. The authors should be consistent in their labeling. Do they mean haplotype here?

Reviewer #3 (Remarks to the Author):

The authors have satisfied this reviewer's requirements by the addition of significant new experiments and by judicious editing of the manuscript.

Below we list point-by-point responses to specific comments made by the referees. Reviewer comments are in black, with our responses in red.

The first part of the paper (identification of that same locus as being the best candidate for a polychromatic greenbeard locus) could be shortened.

We have followed the editor's suggestion and shortened this section by moving associated background information into the introduction.

'Critically, although our analysis identifies two candidate genes that show very tight physical linkage (i.e., they are only ca. 500 bp apart) and functional biochemical coupling and therefore the pair can constitute a single locus (the "tgr locus"), as required by the greenbeard mechanism.' – correct sentence

The sentence was changed to:

Critically, although our analysis identifies two candidate genes, these show very tight physical linkage, as they share a 523 bp common promoter, and are biochemically coupled^{37,38,41}. Hence, the pair can constitute a single locus (the 'tgr locus'), as required by the greenbeard mechanism.

The fact that the phylogenetic trees of both tgr genes do not show identical structures suggests that recombination occurs between both genes and that they do not serve as a single functional locus.

We have also now improved Supplementary Figure 6 to make it clearer that, although the two phylogenetic trees are not 100% identical, the topology only differs in clades where there is very little sequence variation between the strains. Bootstrap values confirm that the topology in these regions is not reliable and more or less random. However, the overall topology (haplotype A and haplotype B) and well-supported subgroups within these groups are indeed identical, which supports the idea that the two genes can be considered as one single locus. The lack of a perfect correlation in the genetic distances does not imply recombination, but merely reflects the fact that the branch lengths will necessarily show random differences because they are determined by stochastic processes. Therefore, a perfect match would only ever be expected in deep time, where you are averaging over enough random events to remove the signature of randomness, but at a local evolutionary scale they will never be perfectly congruent. These points are now also described in the main text.

'Firstly, analyses of these final gene sequences revealed that, while the phylogenetic trees computed from the variants present at these genes are not identical (Supplementary Fig. 6), distances between strains of both tgr genes are very highly positively correlated (Pearson's product moment correlation: $r = 0.89$; $p = 1.84 \times 10^{-128}$; Supplementary Fig. 7A), supporting the idea that the genes show concerted evolution as a single functional locus. Differences in the specific topology of the phylogenetic trees for the two genes only occur where branch lengths are very short and bootstrap values are very small (Supplementary Figure 6, haplotype B), suggesting that there is not enough variability between the sequences of these strains

to infer a reliable phylogeny. However, the overall topologies of the trees are mostly identical (Robinson Foulds distance of 14).'

We have firstly added text describing that these genes share a common promoter, as well as being biochemically coupled, thus further strengthening this argument

Critically, although our analysis identifies two candidate genes, these show very tight physical linkage, as they share a 523 bp common promoter, and are biochemically coupled^{37,38,41}.

'(tgrB1: $r = 0.24$; $p = 0.001$, tgrC1: $r = 0.21$; $p = 0.004$)' – is there a graph/plot showing these results?

Supplementary Figures 7D and E have been added to show these relationships.

Figures

In general the figure captions need better explanations. This is especially true for the supplementary figures.

Changes have been made throughout the figure legends as suggested.

Supplementary Figure 2:

The numbers in parentheses need better explanation.

Numbers in parentheses represent segregation values. This has been added to the figure legend.

2-way and 3-way mixes of strains are shown. In each case, the labelled strain is highlighted bold.

***A.** In pairwise mixes, strain NC28.1 segregates strongly from strains NC105.1 (segregation value = 66.42, see 'measuring segregation' in the methods section for a description of how these were calculated) and NC34.2 (seg = 61.83), while strains NC 34.2 and NC105.1 do not segregate (seg = 16.38). When all three strains are mixed and NC28.1 is labelled, the segregation value is the same as pairwise mixes (seg = 68.11) as it segregates from both other strains. If NC105.1 is labelled, an intermediate segregation value compared to pairwise mixes is observed (seg = 41.64) because it segregates from NC28.1 but not from NC34.2.*

***B.** In pairwise mixes, strain NC28.1 segregates strongly from strains NC105.1 (seg = 66.42) and NC63.2 (seg = 64.61), while strains NC 63.2 and NC105.1 do not segregate (13.48). When all three strains are mixed and NC28.1 is labelled, the segregation value is the same as pairwise mixes (seg = 66.74) as it segregates from both other strains. If NC105.1 is labelled, an intermediate segregation value is observed (seg = 33.70) because it segregates from NC28.1 but not from NC63.2.*

Supplementary Figure 8:

The description says allele A and allele B. In supplementary figure 4C it is stated that TgrB1 and TgrC1 have 18 alleles each. The authors should be consistent in their labeling. Do they mean haplotype here?

Alleles has been changed to haplotype.

Blue dots depict all positions where one strain has a different amino acid than all other strains, black dots depict positions with multiple differences, green dots depict positions with eight strains showing one amino acid (haplotype A), while eleven other strains show a different amino acid (haplotype B).